# Influence of enclosure design on the behaviour and welfare of *Pogona vitticeps*

**Melanie Denommé** [ID]*, **Natalie L. Bakker, Glenn J. Tattersall** [ID]

Department of Biological Sciences, Brock University, St. Catharines, Ontario, Canada

* md19nm@brocku.ca

## Abstract

Complex or naturalistic enclosures have become increasingly accepted as those best-suited to improve an animal's welfare. However, designing such enclosures can be difficult if little is known about the animal in the wild, and enclosures that aim to replicate natural habitats must still be assessed to ensure their assumed benefits are realized. Therefore, this study examined the behaviour and physiology of captive-bred bearded dragons (*Pogona vitticeps*) living in naturalistic- and standard-style enclosures. First, we assessed whether naturalistic-style enclosures better accommodated a lizard's behaviour by examining if lizards in these enclosures were inactive for a similar amount of time as their wild counterparts, if they used their enclosures more evenly than standard-housed lizards, and if naturalistic enclosures provided better thermal heterogeneity than standard enclosures. Then, we examined if living in naturalistic-style enclosures improved the lizard's welfare by examining behaviours related to stress and relaxation as well as heterophil to lymphocyte (H:L) ratios. Although naturalistic enclosures did offer better thermal heterogeneity, evidence that they better accommodated a lizard's behaviour or improved their welfare was equivocal: lizards spent the majority of their day inactive, in one area of the enclosure, and performed similar amounts of behaviours related to stress and relaxation, regardless of enclosure style. Furthermore, H:L ratios were only lower for female lizards in naturalistic enclosures. Our results may have been influenced by the timeline of data collection but could also suggest that standard enclosures are sufficient for *P. vitticeps*, that *P. vitticeps* perceive standard- and naturalistic-style enclosures as similar, or that the potential benefits of naturalistic enclosures were hampered by the enclosure's size. Ultimately, it was apparent that structural complexity alone was insufficient to influence lizard welfare, highlighting the importance of considering the animal's motivations throughout their life and aspects other than enrichment for effective enclosure design.

## Introduction

The popularity of reptiles as pets has grown considerably in the past few decades [1], but this growth has been accompanied by a concomitantly increasing concern for

**Data availability statement:** All files, data, and analyses for this study are available in a public repository on the Borealis dataverse: https://doi.org/10.5683/SP3/GIUPVO.

**Funding:** The data in this article were gathered as a part of M.D.'s PhD thesis. M.D. was supported by a Natural Sciences and Engineering Council (NSERC) Postgraduate Scholarship-Doctoral (PGS D-580167-2023). The data in this article were also gathered as a part of N.L.B.'s NSERC Undergraduate Student Research Award (USRA) (USRA - 592895 - 2024). The research was funded by an NSERC of Canada grant to G.J.T. (RGPIN-2020-05089). The funders had no role in study design, data collection and analysis, decision to publish, or preparation of the manuscript.

**Competing interests:** The authors have declared that no competing interests exist.

these species' suitability and welfare in captivity [2,3]. Concerns about the welfare of pet reptiles primarily stem from our dearth of knowledge about their effective care and husbandry relative to more traditional companion animals [4,5]. The welfare of captive reptiles may be particularly vulnerable to this lack of knowledge; as ectotherms, a reptile's behaviour and physiology are heavily influenced by their environment, of which owners are solely responsible [6,7]. Consequently, to safeguard the health and welfare of reptiles in captivity, research examining the adequacy of husbandry practices is critical.

However, little research exists about reptile behaviour compared to other vertebrates [8,9], and there is a considerable lack of validated welfare indicators for reptiles (i.e., validated by repeated testing with multiple other indicators that should all relate to the same state of welfare; [10]), ultimately making it difficult to determine what husbandry practices may be adequate. When such critical knowledge is absent, it can be more effective to focus on whether or not husbandry practices facilitate the animal's agency. Here, agency is defined as the capacity for animals to engage in voluntary, self-generated, and goal-directed behaviour that they are motivated to perform [11]; to facilitate agency, animals must be able to choose how and where they express their behaviour. Facilitating agency can be effective at promoting welfare for several reasons; the ability to choose can be rewarding itself, animals with agency may be better able to cope with stress, and the provision of choice allows animals to better engage with their environment in ways meaningful to their life history and needs [12–14]. In other words, facilitating agency means animals are better able to perform the behaviours that *they* "want to do" – a critical feature of their welfare [15]. Therefore, if little is known about the animal's needs and motivations, adequate enclosures can be created by focusing on designs that facilitate the animal's agency.

In general, complex enclosures are the most effective at facilitating agency, as they are most likely to provide a variety of opportunities for animals to behave in ways best suited to their motivations [16,17]. Furthermore, complexity that mimics the species' natural environment might be particularly effective at facilitating agency, as this would increase the likelihood that the opportunities provided to animals are relevant to their life histories and potential needs [16,17]. However, naturalness is not a requirement for enclosures to be adequate, as "unnatural" husbandry practices (e.g., training animals, providing them with toys or other artificial objects) can still be effective at facilitating agency and promoting welfare [18,19], and enclosures can be "natural" in ways that are irrelevant to the animal's life history and motivations (e.g., providing trees to fossorial species). Indeed, in reptiles, there is evidence that complex and/or naturalistic environments both improve welfare [20–29] and have no or an inconclusive impact on welfare [30–35].

Therefore, in this article, we will investigate two main hypotheses. First, we will investigate how a complex, naturalistic-style enclosure may or may not accommodate a lizard's behaviour and thereby facilitate their agency compared to a simpler enclosure. Secondly, we will investigate whether lizards in naturalistic enclosures have better welfare than those in standard enclosures. However, because the timing and order by which animals are exposed to enrichment can influence its efficacy (e.g.,

[25,36]), we will investigate each hypothesis at multiple points in time, before and after lizards have experienced both naturalistic and non-naturalistic conditions. Because naturalistic enclosures will include additional furnishings and attempt to mimic aspects of the natural environment, we expect these enclosures will better facilitate a lizard's agency and that the welfare of lizards in these enclosures will be better than those in standard enclosures; we also expect this influence to be evident regardless of the order in which enclosure styles are experienced, and that an enclosure style's influence will increase the longer it is inhabited. In addition, although sex is known to influence some of the outcomes that will be measured [37,38], enclosure style should not influence lizards differently based on their sex; because we expect that naturalistic enclosures will better facilitate agency, the lizard's behaviour should be accommodated and their welfare should be improved regardless of the potentially unique motivations of each sex.

To investigate how naturalistic enclosures may accommodate a lizard's behaviour, we will compare 3 outcomes for lizards in naturalistic and standard enclosures: (1) the amount of time spent inactive, (2) the uniformity of enclosure use, and (3) the thermal heterogeneity of both enclosures. The amount of time lizards spend inactive will be measured in both standard and naturalistic enclosures and also compared to the amount of time that wild *Pogona vitticeps* spend inactive. If naturalistic enclosures effectively provide opportunities to perform a variety behaviours, lizards in these enclosures should spend less time inactive than lizards in standard enclosures and spend similar amounts of time inactive as wild *P. vitticeps*, because wild animals are typically able to perform highly-motivated behaviours [19]. Although *P. vitticeps* are largely inactive in the wild [37,39], too little or too much time inactive could indicate that the enclosure is failing to accommodate their needs or behaviours in some way; for example, lizards may bask excessively if the temperatures provided are inadequate or perform repetitive behaviours incessantly in attempts to escape (see [40]). In either case, large deviations (i.e., extreme values) in the amount of time inactive relative to wild *P. vitticeps* may reflect that the behaviours that captive lizards can perform are restricted and, as a consequence, so too would their agency be restricted [19]. Secondly, lizards in naturalistic enclosures should have either similar or less skewed enclosure use patterns compared to those in standard enclosures. When compared between enclosure styles, areas that are avoided or over-used could indicate that lizards perceive these areas (or the furnishings they include) as aversive, inadequate, or useless. If so, this would limit the opportunities available for lizards to express their behaviour by limiting the effective space in the enclosure, and ultimately would restrict their agency. Finally, to facilitate agency and accommodate lizard's behaviours, naturalistic enclosures should have greater thermal heterogeneity than standard enclosures, as this would improve the lizard's ability to behaviourally thermoregulate [41,42].

In addition to assessing if naturalistic enclosures better accommodate a lizard's behaviour, the welfare of lizards in naturalistic enclosures and standard enclosures will be investigated by comparing: (1) the amount of time spent inactive with legs stretched (ILS), (2) the amount of time the lizards spend performing repetitive interactions with the barriers (IWB), and (3) the heterophil to lymphocyte (H:L) ratios of lizards in both enclosures. The ILS posture will be used as an indication of relaxation, and therefore positive welfare for the following reasons: this posture is only adopted when the lizard is asleep or after they have been inactive for some time, lizards exit out of the posture when they observe a potential threat (i.e., a human), the posture does not appear to confer any survival benefit (e.g., facilitate thermoregulation), and the posture requires that a lizard's palms or soles are not in contact with the substrate, potentially making them more vulnerable to predation. Essentially, ILS may be indicative of positive welfare states because it is only observed in "opportunity situations" when the potential "cost" of performing such a behaviour is low [43]. Therefore, we expect lizards in naturalistic enclosures to perform ILS longer than lizards in standard enclosures. As research has previously shown [40], IWB will be considered as an escape attempt and therefore an indication of frustration or stress; consequently, if lizards in naturalistic enclosures have better welfare than those in standard ones, this behaviour should be performed less in naturalistic enclosures. Finally, H:L ratios will be used to infer chronic stress. Heterophils and lymphocytes are white blood cells that respond to infections and diseases, but their levels in the blood are also influenced by the hypothalamic-pituitary-adrenal (HPA) axis and glucocorticoids; specifically, the number of heterophils can increase and the number of lymphocytes can

decrease when the animal experiences some stressor [44,45]. For these reasons, H:L ratios have been positively cor-related with repeated and long-term stressors (i.e., low social rank, migration in passerines) as well as disease or infec-tion, but they are unresponsive to short periods of acute stress (i.e., handling), making them a useful indicator of chronic stress [44]. Furthermore, an influence of enclosure style on H:L ratios has been observed in reptiles; in turtles, H:L ratios were lower (indicating lower chronic stress) when housed in naturalistic compared to barren enclosures [22]. Therefore, if naturalistic enclosures promote welfare, we expect lizards in these enclosures to have lower H:L ratios compared to lizards in standard enclosures.

## Methods

### Ethical note

All procedures were approved by Brock University's Animal Care Committee (AUP 21–08–02). Behavioural data and data regarding enclosure temperatures were collected within normal daily routines that the lizard experienced, causing little to no additional stress. Blood samples were taken by competent animal care personnel trained by a veterinarian.

### Animals and husbandry

As one of the most popular pet lizards in captivity [1] and a useful model organism for some research [46], *Pogona vit-ticeps* will be used to examine how a lizard's agency and welfare may be influenced by enclosure design. These lizards are native to the semi-arid woodlands of central Australia and are one of several *Pogona* species found throughout the country [47]. In the wild, *P. vitticeps* spend much of the day inactive [37,39], and are often found on the ground or on ele-vated perches [48,49]; consequently, *P. vitticeps* are typically considered to be semi-arboreal [47,50]. During the breeding season, males may use elevated perches to defend territories [37,47], but a definitive home range size is unknown for *P. vitticeps* [37]. Furthermore, as female *P. vitticeps* dig burrows prior to oviposition [47,50–52], digging is also considered to be an important part of their behavioural repertoire.

Observations were collected on 24 bearded dragons (*P. vitticeps*) over 3 years. Animals were obtained from 2 separate breeders; 20 lizards came from one clutch, and 4 from another clutch. Lizards entered laboratory conditions September 31st, 2021, when they were approximately 2 months old. At 1 year old, around the onset of sexual maturity (i.e., between 1–2 years old [52]), lizards were sexed using external anatomy and morphological differences. The population consisted of 11 males and 13 females.

Lizards were housed individually in black PVC cages (60 cm x 61 cm x 40.6 cm; L x W x H, approximately 39.2 gal-lons). The front of the cage had a clear acrylic door 50.8 cm long and 25.4 cm high. This door had latches at the top of the plexiglass that could rotate, and the door swung downwards to open. In the back right corner of all enclosures, a 30 cm-by-30 cm tile was securely wedged on an angle to create a basking spot and warm hiding spot directly underneath. Above this tile, a 40-watt incandescent bulb and 13-watt UVB-150 Exo Terra® compact fluorescent bulb were mounted to provide heat and UV radiation in the approximate center of the tile. Each day, the maximum temperature of this tile was checked using a FLIR TG165 Spot Thermal Camera (FLIR Systems, Inc., Wilsonville, OR. U. S. A.) in a random enclosure to ensure that the maximum (basking) temperature was not below 35°C or above 50°C. The front of the enclosure always approximated room temperature, between 21–24°C. The UV bulbs of all cages were checked every few months to ensure UVB intensity was maintained using Solarmeter® Digital Ultraviolet Radiometer (model 6.2 for UVB). Humidity was also checked daily in a random enclosure to ensure levels between 25–90% relative humidity immediately on top of the sub-strate. The photoperiod in the room was set to 12L:12D and did not change throughout the course of the year. However, the room in which the lizards were housed experienced fluctuations in humidity annually. One-hour shifts in the timing of the photoperiod due to daylight savings occurred twice per year.

Each day, lizards were provided fresh water and a salad of chopped vegetables (typically dandelion greens and cilan-tro or parsley), butternut squash, and Mazuri ® Insectivore Diet. As juveniles (i.e., less than 1 year old) live insects were

provided daily. As adults, live insects were provided approximately 2-3x per week. Insects provided were typically crickets (*Acheta domestica*), but occasionally Ivory Head cockroaches (*Eublaberus sp.*) or mealworms (larvae of *Tenebrio molitor*) were provided. Insects were always dusted with a variety of powdered supplements; either calcium powder fortified with vitamin D, without vitamin D, with multivitamin powder, or with some combination of these three. When lizards were not fed insects, powdered supplements were sprinkled onto the salad.

All lizards were weighed and were checked for health weekly. Standard enclosures were fully cleaned (i.e., substrate replaced, floor and parts of the walls washed with soap and water) every two weeks while naturalistic enclosures were fully cleaned every three weeks. The difference in enclosure cleaning frequency was due to the labour required to change cages and the rate at which the enclosures became soiled; standard enclosures tended to soil quicker than naturalistic enclosures. A description of each enclosure style can be found below.

## Enclosure styles

On January 18th, 2022, 13 lizards were placed into naturalistic enclosures and 11 into standard enclosures. After 200 days, on August 5th, 2022, lizards in naturalistic enclosures were swapped to standard enclosures and vice versa. Therefore, all lizards experienced both enclosure styles, but at different time points and in different orders. For their health, one lizard was swapped from a standard enclosure to a naturalistic style on April 25th, 2024.

Standard enclosures included paper substrate and a paper hide (Bio-Huts for Rats™ from Bio-Serv) in the front left corner of the enclosure. Naturalistic enclosures had a loose substrate of 70:30 coco peat: play sand mixture, an Exo Terra® Reptile Cave Hideout (hereafter called "naturalistic hide"), a piece of cork bark, and the same paper hide. In naturalistic enclosures, the naturalistic hide was placed in the front left corner, the paper hide in the back left corner, and the cork bark was placed on top of the paper hide at an angle. This created several extra hiding spots, allowed the creation of dens, and provided a climbing opportunity (Fig 1).

We chose to label the "naturalistic" enclosure as naturalistic for several reasons. Primarily, "naturalistic" best described the enclosure's appearance compared to the standard enclosures and also captured the intended outcome of the modifications in the naturalistic enclosure (i.e., to simulate aspects of the natural environment and promote natural behaviour patterns). The label "complex" was avoided as it would have shifted the focus away from this rationale; although naturalistic enclosures included 2 additional furnishings (cork bark with paper hide underneath), 2 other furnishings were *modified* (substrate and naturalistic hide in front left corner). The label "enriched" may also have been misleading. In the literature, "enriched" will typically (and should exclusively; [53,54]) refer to modifications that demonstrably improve the animal's welfare in some way [55]. As the aim of this research was to investigate this issue, and consequently we had no evidence that the resources we provided to lizards improved their well-being beyond basic husbandry requirements, the label

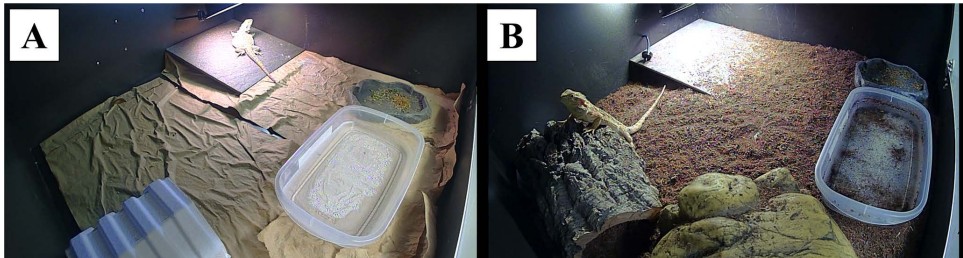

**Fig 1. Images of standard (A) and naturalistic (B) enclosures in which *Pogona vitticeps* were kept.** Standard enclosures included a paper substrate and paper hide (visible in the front left). Naturalistic enclosures included a loose substrate, cork bark (visible in the back left), paper hide (not visible; used to prop up cork bark) and a naturalistic hide (visible in front left). Both enclosure styles included a ceramic tile, food bowls, and water bowls. Images were taken from video recordings by TLC200 Pro Brinno cameras and therefore have a minor fisheye-lens distortion.

"enriched" was avoided. This is also why the items in the enclosures are referred to as "resources" or "furnishings" rather than "enrichments" (a similar convention has been used in other literature; [56]).

Furthermore, enclosure designs were chosen to be broadly applicable. Reptiles – especially *P. vitticeps* – are increasingly popular pets [1,57], and in the pet trade, a hide and a warm/basking spot is almost always provided at minimum (representing the "standard" enclosure) [58–61]. Loose substrates, additional hiding spots, and climbing structures (representing the "naturalistic" enclosure) are less common, though this depends on the species [58–61]. In addition, *P. vitticeps* has recently been recognized as a useful model organism for certain research [46]. Therefore, as enclosures were maintained by relatively few people and approved by our local Animal Care Committee, both standard and naturalistic enclosures may represent designs that are feasible in a research environment.

## Outcomes measured

**Behaviours.** Observations of behaviour were always collected using timelapse cameras (Brinno TLC200 Pro) set to capture 1 frame every 2 seconds, and always included the entire 12-hour lights-on period. Cameras were mounted in the front left corner on a metal stand in each lizard's cage; while this position maximized the view of the enclosure, a small blind spot directly underneath the camera was unavoidable (see S1 Appendix). Behavioural observations took place between February 2022 and October 2023 (Table 1). The analysis of videos to measure behaviour was completed using the BORIS software [62]. At each period of observation, the lizard's behaviours were scored continuously. Definitions and additional details about the behaviours scored are provided below.

Inactivity was defined as any instance when the lizard did not move for at least 3 consecutive seconds. Bouts of inactivity were not interrupted by changes in posture or slight shifts of the body. Inactivity ceased if the lizard moved a distance greater than or equal to half their approximate snout-vent-length (i.e., the length from the tip of their nose to the midsection of their torso) or if they began to perform some other behaviour, regardless of whether or not this behaviour involved large movement (e.g., beard stretching, pooping, etc.). Because hiding spots in the enclosure were too small to easily allow other behaviours or significant movement, any time spent hiding (i.e., using a furnishing to be out of sight of an observer looking in the front of the enclosure) was also considered time spent inactive.

Inactive and leg stretched (ILS) was defined as inactivity whilst at least 1 limb was either partially or fully extended (i.e., the knee/elbow was mostly or completely straight), with the palm or sole facing away from the substrate. In this posture, the pelvis was always in full contact with the substrate. At least 1 stretched limb as described above had to be visible for this behaviour to be recorded, but other parts of the body could have been obscured from view. This behaviour ended once no limbs were sufficiently extended; this could occur due to movement or the lizard starting to perform some other behaviour.

**Table 1. Timeline of data collection.**

| Month and year | Data collected | Lizard age (days) | Time in an enclosure style (days) |
|---|---|---|---|
| June/July 2022 | Behavioural observation – Naïve | 308–333 | 148–173 |
| August 2022 | Behavioural observation – Swap | 361–367 | 1–7 |
| January 2023 | Behavioural observation – Experienced | 532–536 | 173–177 |
| September/October 2023 | Behavioural observation – Long-term | 777–783 | 418–424 |
| April – June 2024 | Blood collection | 968–1044 | 609–685 |
| May 2024 | Thermal images of enclosures taken | | |

Twenty-four *Pogona vitticeps* lizards entered the lab on September 30th, 2021, and were placed into either a naturalistic or standard style enclosure on January 18th, 2022. The enclosure style of all lizards was swapped on August 5th, 2022. A round of behavioural observations consisted of recording all 24 lizards for the entire 12-hour lights-on period and scoring inactivity, inactive and leg stretched (ILS), repetitive interactions with barriers (IWB), and enclosure use. Blood collections occurred sporadically, at 7 different times, between April and June 2024.

Interacting with barriers (IWB) was defined as walking into or attempting to climb up a barrier (i.e., one of the walls of their enclosure) for at least 90 consecutive seconds or 3 repetitions (i.e., when the lizard had gone from one corner of a wall to another 3 times in a row). This behaviour ceased if the lizard performed any other behaviour or if they paused and failed to continue interacting with barriers within 90 seconds.

To measure enclosure use, enclosures were divided into 4 approximately equal quadrants, based on the dimensions of the tile (Fig 2). The tile was effective at demarcating the quadrants, as its dimensions were almost exactly ¼ the dimensions of the enclosure's floor. Lizards were considered to have crossed into a quadrant when the majority of their head was within that quadrant. The head was chosen as it could be easily delineated from the rest of the body and lizards almost always moved in the direction their head was pointing; therefore, this location was considered to represent their "intended" location.

**Thermal heterogeneity.** To assess thermal heterogeneity, thermal images were taken of all enclosures in May 2024 around 0900, 1100, and 1500. At this point, there were 12 naturalistic and 12 standard style enclosures because one lizard had swapped into a naturalistic enclosure one month prior. Images were collected using a FLIR T1030SC thermal camera with a FLIR OSX Precision Optical System lens (FLIR Systems). Temperatures were extracted from thermal images using the image processing software FIJI [63] and ThermImageJ [64]. By using radiometric data stored in each pixel, appropriate emissivity (0.95), distance (1 m), and microclimate (air temperature and relative humidity), ThermImageJ estimates the surface temperature of objects in the pictures taken, using the same algorithms employed by FLIR. To examine substrate temperatures, a region of interest was drawn which encompassed the substrate, but excluded the furnishings (e.g., tile, cork bark), lizard, and the area at the front of the enclosure with the food and water bowl (Fig 3). Within this region of interest, ThermImageJ provides information about the maximum, minimum, average, and standard deviation of the temperatures. A region of interest was also drawn around the tile, but the lizard was excluded if necessary. This was used to corroborate that tile temperatures (i.e., basking temperatures) were within acceptable ranges, in addition to the daily checks that were already being performed.

**Heterophil to lymphocyte (H:L) ratios.** Collection of blood samples took place periodically between April 2024 and June 2024. The lizard whose enclosure style was swapped in April 2024 was excluded from these analyses to control for the influence that time in an enclosure style could have on H:L ratios. Samples were always collected in the morning, between 0900–1100. For sample collection, lizards were gently restrained, and blood was taken from the ventral coccygeal vein using a 25-gauge needle then immediately stored in tubes with EDTA. Blood smears were prepared within 24 hours of collection. If a lizard had to be sampled more than once (e.g., due to issues with preparing blood smears), at least 28 days elapsed between sample collection, and no lizards were sampled more than 3 times (for more details, see S2 Appendix).

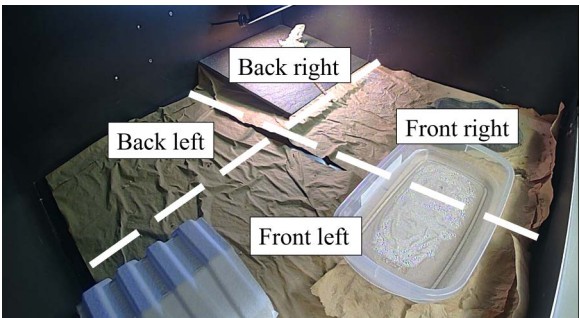

**Fig 2. Demarcation of the 4 quadrants used to assess enclosure use in *Pogona vitticeps*.** In this figure, a standard-style enclosure is seen. Quadrants were based off the dimensions of the tile, which was almost exactly as large as ¼ of the area of the enclosure. A tile and basking lamp are present in the back right quadrant, a food bowl is present in the front right quadrant, a water bowl spans the front left and right quadrants, a paper hide is present in the front left quadrant, and the back left quadrant is empty.

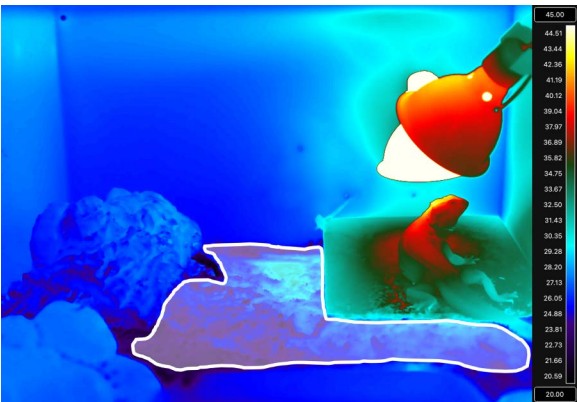

**Fig 3. Thermal image with the area that was used to assess substrate temperature highlighted in white.** In this figure, a naturalistic-style enclosure is seen. This region of interest excluded the tile, the lizard, any furnishings (i.e., cork bark, naturalistic hide) and the area at the front of the enclosure with the food and water bowl.

To prepare blood smears, a small volume of blood was collected with a microcapillary tube lined with heparin and 1–2 drops of blood were placed onto a microscope slide. The smear was created using the push-slide technique and samples were then quickly dried using cold air from a hair dryer. Smears were allowed at least 10 additional minutes to air-dry undisturbed before staining with Wright-Geisma stain (VWR ® North America; Catalogue #: 10143–106). Approximately 80 µL of stain was applied to the smear and allowed to dry for 2 minutes. Then, 150 µL of deionized water was added to the smear and allowed to sit for 1 minute. After this, the same amount of water was applied again and allowed to sit for 1 minute. Finally, the smear was gently but thoroughly rinsed with 1000 µL of deionized water 1–3 times and allowed to air-dry for at least 40 minutes.

H:L ratios were determined by following established protocols [44,65]. Therefore, blood smears were observed under 1000X magnification with the aid of immersion oil and the first 100 leukocytes encountered in the smear were classified as lymphocytes, heterophils, monocytes, basophils, or azurophils based on cell morphology (i.e., [66–68]). Then, H:L ratios were calculated by dividing the number of heterophils by the number of lymphocytes observed. The identification of leukocytes was done by an observer blind to the lizard's identity, sex, and enclosure style.

## Data analysis

***General methods for all analyses.*** All analyses were performed in R version 4.3.2 [69]. If data could not be effectively analyzed by models created using the glm, glmer, or lmer functions from the lme4 package [70] – typically due to a high number of zeros or skewed data – analysis was performed instead using generalised linear models created using the gamlss function from the gamlss package [71]. This function provided access to a greater variety of distributions to better model non-normal data. For example, zero-adjusted distributions allow data with zeros to be analyzed by distributions that would typically be unable to tolerate zeros (e.g., gamma or beta distributions). To do this, zero-adjusted distributions use a hurdle model to analyze data (i.e., analysis is done in two parts); first, the probability of a zero is modelled using a binary model, then all non-zero values are analyzed with the original distribution [72].

The fit of all models was assessed using plots of model residuals. However, plots of model residuals were created with different functions in R depending on the model; models created with the gamlss function were assessed using the plot and wp functions from the gamlss package, whereas models created with the glm, lmer, or glmer function were assessed using the simulateResiduals function from the DHARMa package [73]. For brevity, the result of these assessments will only be reported if issues with model fit were evident.

Because sex could influence the outcomes measured [37,38], it was included during analysis; however, we did not have *a priori* reasons to expect that sex would influence the potential effects of enclosure style. Likelihood ratio tests were used to identify which factors (e.g., enclosure style, sex, etc.) in what combination (e.g., interaction, additive, separate) most influenced model fits and thereby identify which models would be further investigated. First, likelihood ratio tests were used to compare a model with an interaction between factors (e.g., enclosure style * sex) against a model with only their main effects present (e.g., enclosure style + sex). The function used to perform this test depended on how the model was created: likelihood ratio tests were performed with the lrtest function from the VGAM package [74] for models created with glm, the Anova function from the car package [75] was used for models created using glmer or lmer, and the LR.test function from the gamlss package was used for models created with gamlss. If including the interaction did not significantly influence model fit, likelihood ratio tests were used to compare the main effects models against models which included each factor alone (e.g., enclosure style OR sex). Again, the function used for this depended on how the model was created: the function drop1 from the stats package was used for models created with glm, the Anova function from the car package was used for models created with lmer or glmer, and the function drop1All from the gamlss package was used for models created with gamlss. Where necessary, model estimates, contrasts, and p-values adjusted with the Bonferroni method were determined using the avg_predictions or avg_slopes function from marginaleffects package [76] or the emmeans function from the emmeans package [77].

**Behavioural analyses.** To account for the influence of age and time in an enclosure style [25,36] each behaviour (inactivity, ILS, IWB, and enclosure use) was analyzed separately at 4 distinct points in time (Table 1). Analysis of inactivity and IWB utilized percentages that were based on the time the lizard's lights were on subtracted by any time the lizard spent in the camera's blind spot; in other words, percentages are the percent time the lizard's behaviour was known to the observer. Typically, lizards spent most of their time outside of the blind spot, but there were 9 instances across all time points where a lizard spent 30% or more of their lights-on period in the blind spot (see S1 Appendix). If a lizard remained in the blind spot without interruption for an extended period (e.g., > 3hrs), this was typically because they were curled around the base of the camera and completely inactive; however, the same amount of time could also be accumulated by multiple short bouts in the blind spot, during which a lizard was less likely to be inactive. Therefore, time spent in the blind spot was subtracted from the total 12-hour lights-on period to avoid making assumptions about a lizards' behaviour whilst accounting for the time spent within the blind spot.

The percent of time spent inactive during the lights-on period within the camera's view (including the time spent inactive and with their legs stretched) was summed with the percent of the time spent hiding during the lights-on period within the camera's view; hereon, this will be called the percent of known time inactive. The percent of known time inactive was compared between enclosure styles and sexes at each of the 4 time points using a generalized linear model. To account for the unique qualities of percent data, percentages were divided by 100 and a beta distribution with a logit link was used. To compare the percent of known time inactive between wild *Pogona vitticeps* and captive lizards, a one-sample Wilcoxon test was used. Based on recent publications, the typical percent of the daytime that wild *P. vitticeps* spend inactive was considered to be 80% [37]. This test was performed for each enclosure style at each of the 4 observation points.

Initially, we aimed to compare the percent time lizards spent inactive and with stretched legs (ILS) between enclosure styles; unfortunately, after videos were analyzed, it became evident that ILS was performed relatively few times and typically for short durations, rendering this analysis ineffective. Instead, the number of lizards that performed ILS for at least 5% of the day was compared between enclosure styles using a logistic regression. The logistic regression was created using the glm function from the stats package and utilized a binomial distribution with a logit link, and separate models were created for each of the 4 time periods. Odds ratios were calculated using the tab_model function from the sjPlot package [78].

The percent time lizards spent performing IWB (based on the time the lights were on and lizards were not within the blind spot) was compared between enclosure styles at the 4 distinct time points using generalized linear models. As

before, percentages were converted to proportions for analysis, but because IWB was relatively rare, a 0- and 1-adjusted beta distribution with a logit link was used.

To examine enclosure use between enclosure styles, the proportion of time a lizard spent in each quadrant was calculated. This proportion was based on the total amount of time in all 4 quadrants and was not influenced by the camera's blind spot, as a lizard within the blind spot could only be in the front left quadrant. In addition, the influence of sex was not included in these analyses for two reasons: (1) we did not expect that sex would significantly influence the uniformity of enclosure use, and (2) because models for these analyses must include an interaction between quadrant and enclosure style, and the sample size was insufficient to include a second interaction. The front right quadrant was chosen as the reference level for models, as this quadrant contained the food bowl and therefore should be occupied similarly regardless of enclosure style. Data were analyzed using generalized linear mixed models and used a 0- and 1-adjusted beta distribution with a logit link to accommodate zeros in the data. A random intercept was included for the lizard's identity, the proportion of time in a quadrant was the response variable, and an interaction between the quadrant and the lizard's enclosure style were the predictors. Because gamlss models with random effects may have conservative estimates of standard error [79] the alpha for these analyses was adjusted to 0.025 and 97.5% confidence intervals were used for figures of model estimates.

**Thermal heterogeneity.** The standard deviation of temperatures recorded in the substrate was used to examine thermal heterogeneity between enclosure styles. Linear mixed-effects models with a gaussian distribution were used for analysis. These models included a random intercept for the enclosure's identity, as multiple measurements were taken for each enclosure. An interaction between enclosure style and the time the image was taken was also considered, as the time at which images were taken could influence the temperature of the enclosure.

**Heterophil to lymphocyte (H:L) ratios.** Heterophil to lymphocyte (H:L) ratios were analyzed using generalized linear models with a gamma distribution and a log-link function to accommodate the slight right-skew of data. The influence of sex on H:L ratios was considered in this analysis as a relationship between white blood cell counts, corticosterone concentrations (which can influence H:L ratios; [44]) and sex is not clear in *Pogona* species. In wild *P. vitticeps*, only basophil counts have been influenced by sex [38], but such haematological parameters can differ greatly in captive populations [80] and no similar investigation has been performed with captive *P. vitticeps*; in addition, corticosterone concentrations can differ greatly depending on sex and season in wild *P. barbata* [81].

## Results

### Behavioural analyses

*Inactivity.* Predictions of the duration of time spent inactive were similar when models included an interaction between enclosure style and sex compared to models which included only the additive effects of these factors at the naïve time point ($\chi^2_{df=1}=0.69$, $p=0.41$), the swap time point ($\chi^2_{df=1}=0.03$, $p=0.87$), the experienced time point ($\chi^2_{df=1}=1.44$, $p=0.23$) or the long-term time point ($\chi^2_{df=1}=1.37$, $p=0.24$). The amount of time spent inactive was also not differently predicted by models that included only the influence of enclosure style (Naïve: $\chi^2_{df=1}=0.46$, $p=0.50$; Experienced: $\chi^2_{df=1}=0.34$, $p=0.56$; Long-term: $\chi^2_{df=1}=0.82$ $p=0.37$) or only the influence of sex (Naïve: $\chi^2_{df=1}=0.07$, $p=0.79$; Experienced: $\chi^2_{df=1}=2.57$, $p=0.11$; Long-term: $\chi^2_{df=1}=0.51$ $p=0.48$). However, at the swap time point, inactivity was better predicted by a model that included both enclosure style and sex compared to a model which included only enclosure style ($\chi^2_{df=1}=4.21$, $p=0.04$) or only sex ($\chi^2_{df=1}=5.22$, $p=0.02$). At this time point, lizards in standard-style enclosures were inactive for longer durations (Untransformed model estimate ± SE: $\beta=2.39\pm0.18$) compared to lizards in naturalistic enclosures ($\beta=1.86\pm0.17$, $p=0.04$; Fig 4B). Furthermore, male lizards were inactive for longer ($\beta=2.47\pm0.20$) than females ($\beta=1.87\pm0.15$, $p=0.03$; Fig 4F) at the swap time point.

One-sample Wilcoxon tests also found that, at each time point and for both enclosure styles, lizards were inactive for more than 80% of their day (all time points and groups $p<0.001$); indeed, the percent of known time inactive observed was almost always greater than 80% for all groups and time points (Fig 4).

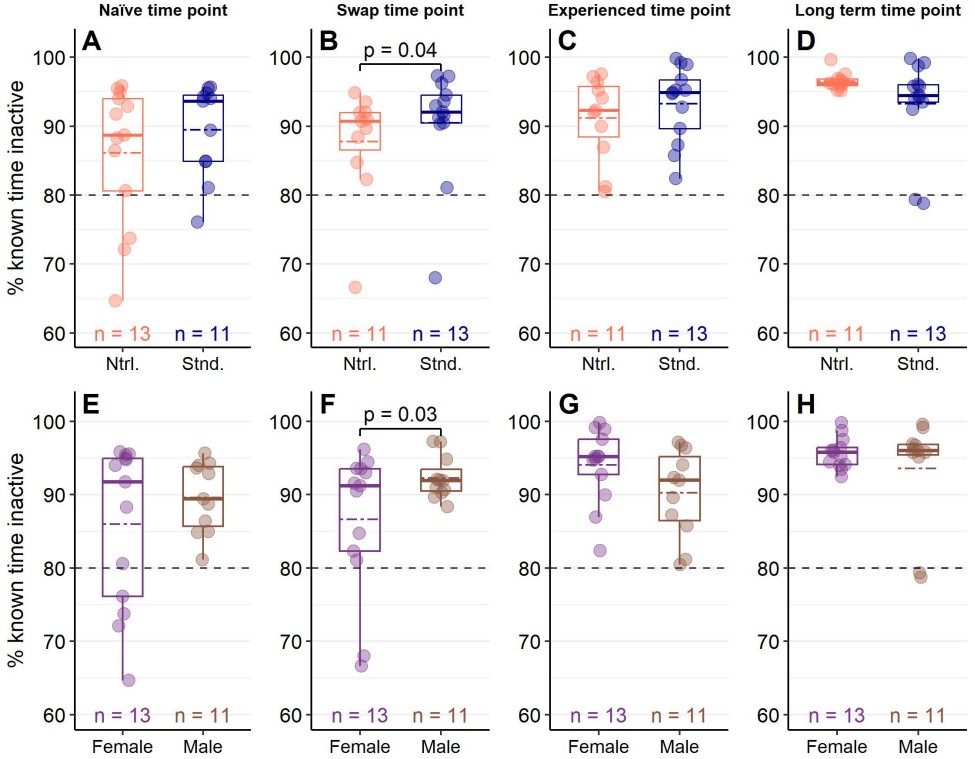

**Fig 4. Percent time inactive in Pogona vitticeps was mostly independent of enclosure design or sex.** Percent of known time *P. vitticeps* were inactive in either naturalistic or standard enclosures (A – D) or between female and male lizards (E – F). Lightly-coloured dots represent individual values and colour-coded dashed lines represent mean values. A black dashed line at 80% is provided as a reference to the amount of inactivity typical of wild *P. vitticeps* estimated from [37]. Data in plots (A) and (E) were collected when lizards were sub-adults and had been in their enclosure style for at least 148 days (Naïve time point). Data in plots (B) and (F) were collected when lizards were adults (~365 days old) within a week of their swap to the opposite enclosure style (Swap time point). Data in plots (C) and (G) were collected when lizards were around 530 days old and had been in their enclosure style for at least 170 days (Experienced time point). Data in plots (D) and (F) were collected when lizards were over 2 years old and had been in their enclosure style for at least 418 days (Long-term response point). Timelapse cameras were used to record behaviour, and percent of known time inactive represents the amount of time lizards spent either immobile or hiding within the camera's view. Wilcoxon one-sample tests found that inactivity was always greater than 80% ($p < 0.001$ for all groups). Generalized linear models found that inactivity only differed between enclosure styles ($p = 0.04$; B) and sexes at the swap time point ($p = 0.03$; F).

**Inactive and stretched leg.** A logistic regression was created to compare the number of lizards that were inactive and had a leg stretched for at least 5% of their day between enclosure styles and sexes at each point in time. At the naïve time point, model estimates were not accurately predicted when an interaction between enclosure style and sex was included; therefore, this model was excluded. Otherwise, model predictions of the prevalence of ILS were similar between models that included an interaction between enclosure style and sex and models that included only the additive effects of both factors at the swap time point ($\chi^2_{df=1} = 1.32$, $p = 0.25$), experienced time point ($\chi^2_{df=1} = 3.30$, $p = 0.07$), and long-term time point ($\chi^2_{df=1} = 3.11$, $p = 0.08$). In addition, at all time points, the inclusion of enclosure style alone (Naïve: $\chi^2_{df=1} = 0.26$, $p = 0.61$; Swap: $\chi^2_{df=1} = 0.12$, $p = 0.73$; Experienced: $\chi^2_{df=1} = 0.35$, $p = 0.55$; Long-term: $\chi^2_{df=1} = 0.68$, $p = 0.41$) or sex alone (Naïve: $\chi^2_{df=1} = 0.26$, $p = 0.61$; Swap: $\chi^2_{df=1} = 0.12$, $p = 0.73$; Experienced: $\chi^2_{df=1} = 3.24$, $p = 0.07$; Long-term: $\chi^2_{df=1} = 0.14$, $p = 0.71$) failed to influence model predictions of the prevalence of ILS compared to null models (Fig 5).

**Interacting with barriers (IWB).** Including an interaction between enclosure style and sex did not influence predictions of the duration of IWB compared to models that included only the additive effects of both factors at the naïve time point ($\chi^2_{df=1} = 0.16$, $p = 0.69$), swap time point ($\chi^2_{df=1} = 0.11$, $p = 0.74$), and the experienced time point ($\chi^2_{df=1} = 0.002$, $p = 0.97$). At the

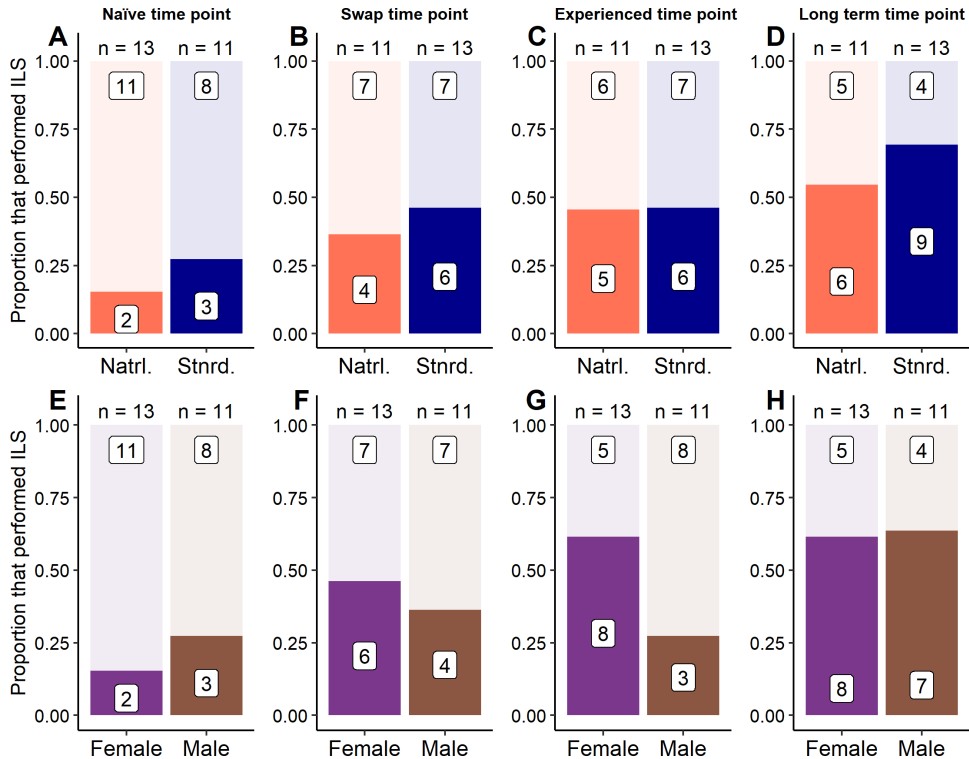

**Fig 5. Performance of behaviours related to relaxation independent of enclosure design and sex in Pogona vitticeps.** Proportion of *P. vitticeps* that were inactive and leg stretched (ILS) for at least 5% of their observed day between naturalistic and standard enclosures (plots A – D) or between female and male lizards (plots E – H) at 4 different points in time. White boxes with numbers inside of coloured-in or lightly-coloured areas provide the count of lizards that either performed ILS for at least 5% of their day (coloured-in area) or did not (lightly-coloured area). Numbers above bars in each plot provide the total number of lizards in each group. Data in plots (A) and (E) were collected when lizards were sub-adults and had been in their enclosure style for at least 148 days (Naïve time point). Data in plots (B) and (F) were collected when lizards were adults (~365 days old) within a week of their swap to the opposite enclosure style (Swap time point). Data in plots (C) and (G) were collected when lizards were around 530 days old and had been in their enclosure style for at least 170 days (Experienced time point). Data in plots (D) and (F) were collected when lizards were over 2 years old and had been in their enclosure style for at least 418 days (Long-term response point). A logistic regression was performed to compare the number of lizards performing ILS between groups but found no influence of sex or enclosure style at any point in time ($p > 0.05$).

long-term time point, and interaction between enclosure style and sex could not accurately calculate model estimates and was therefore excluded from this comparison. At the naïve time point ($\chi^2_{df=1} = 0.56$, $p = 0.46$), swap time point ($\chi^2_{df=1} = 2.55$, $p = 0.11$), experienced time point ($\chi^2_{df=1} = 0.17$, $p = 0.68$) and long-term time point ($\chi^2_{df=1} = 0.37$, $p = 0.55$), the inclusion of only enclosure style failed to influence predictions of IWB durations. Similarly, sex failed to influence model predictions of the duration of IWB at the naïve time point ($\chi^2_{df=1} = 3.33$, $p = 0.07$), experienced time point ($\chi^2_{df=1} = 0.23$, $p = 0.63$), and long-term time point ($\chi^2_{df=1} = 3.27$, $p = 0.07$). However, at the swap time point, including sex did influence IWB predictions ($\chi^2_{df=1} = 7.38$, $p = 0.007$); a model which included only the influence of sex found that IWB was performed for shorter durations by males (Untransformed model estimate ± SE: $\beta_{Male} = -3.52 \pm 0.42$) compared to females ($\beta_{Female} = -2.57 \pm 0.26$, $p = 0.04$; Fig 6).

**Enclosure use.** Model predictions that included an interaction between quadrant and enclosure style did not differ from models that included only the additive effects of both factors at the naïve time point ($\chi^2_{df=3} = 4.11$, $p = 0.25$), swap time point ($\chi^2_{df=3} = 4.93$, $p = 0.18$) or the experienced time point ($\chi^2_{df=3} = 2.20$, $p = 0.53$). However, at the long-term time point, an interaction between quadrant and enclosure style improved model fit ($\chi^2_{df=3} = 8.67$, $p = 0.03$). Regardless, the time spent in each quadrant was not influenced by the lizard's enclosure style for this model at this time point ($p > 0.025$). Similarly, enclosure style failed to influence model fit at the naïve time point ($\chi^2_{df=1} = 0.02$, $p = 0.89$), the swap time point ($\chi^2_{df=1} = 0.014$,

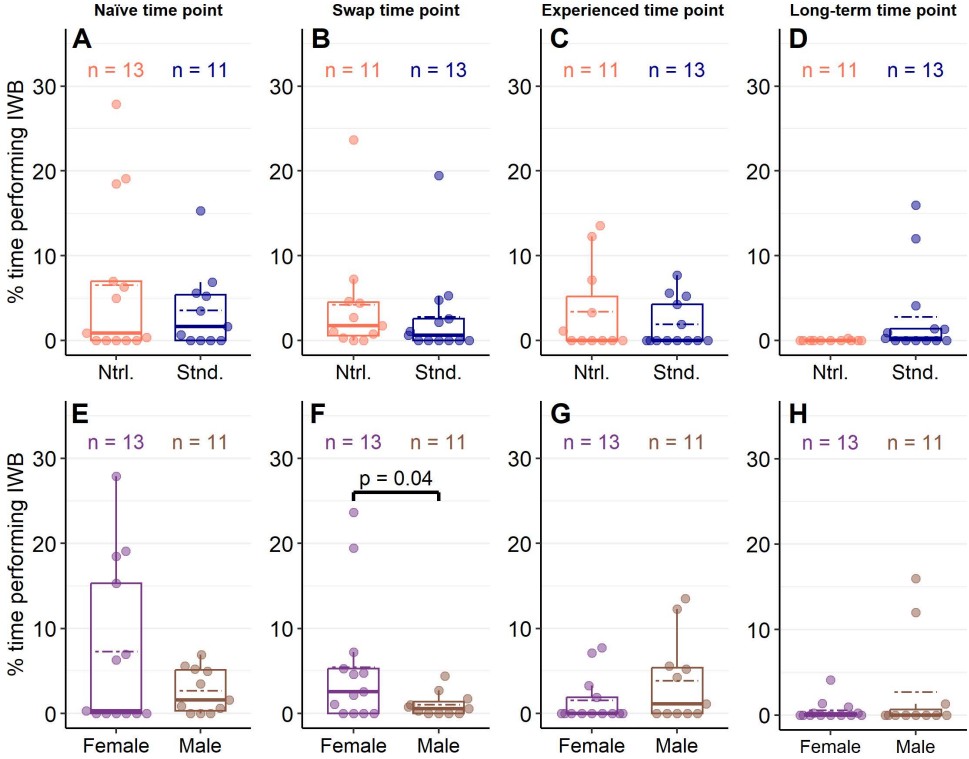

**Fig 6. Performance of behaviours related to stress unaffected by enclosure design and mostly independent of sex.** Percent of known time *Pogona vitticeps* interacted with barriers (IWB) in either naturalistic or standard enclosures (A – D) or between female and male lizards (E – F) at 4 different time points. Lighter-coloured dots represent individual values and colour-coded dashed lines represent mean values. Data in plots (A) and (E) were collected when lizards were sub-adults and had been in their enclosure style for at least 148 days (Naïve time point). Data in plots (B) and (F) were collected when lizards were adults (~365 days old) within a week of their swap to the opposite enclosure style (Swap time point). Data in plots (C) and (G) were collected when lizards were around 530 days old and had been in their enclosure style for at least 170 days (Experienced time point). Data in plots (D) and (F) were collected when lizards were over 2 years old and had been in their enclosure style for at least 418 days (Long-term response point). Generalized linear models were used to compare data between groups within each plot; only an influence of sex was detected at the swap time point (F). At this time point (F), compared to female lizards, male lizards performed IWB for less time ($p = 0.04$).

$p = 0.91$), and the experienced time point ($\chi^2_{df=1} = 0.42$, $p = 0.52$). Enclosure use was only influenced by quadrant (main effect alone) at the naïve time point ($\chi^2_{df=3} = 149.83$, $p < 0.001$), the swap time point ($\chi^2_{df=3} = 104.50$, $p < 0.001$) and the experienced time point ($\chi^2_{df=3} = 53.88$, $p < 0.001$). At all time points, the pattern of enclosure use was the same; compared to the front right quadrant, lizards spent more time in the back right quadrant at the naïve time point (Untransformed model estimate ± standard error: $\beta_{Back\ right} = 3.88 \pm 0.25$, $p < 0.001$), the swap time point ($\beta_{Back\ right} = 3.33 \pm 0.29$, $p < 0.001$), the experienced time point ($\beta_{Back\ right} = 2.37 \pm 0.32$, $p < 0.001$), and the long term time point ($\beta_{Back\ right,\ Naturalistic} = 2.96 \pm 0.47$, $p < 0.001$; $\beta_{Back\ right,\ Standard} = 2.27 \pm 0.47$, $p < 0.001$). Indeed, it is evident that lizards spent the majority of their time in the back right quadrant compared to all other quadrants, and the time spent in all other quadrants was similar (Fig 7).

**Thermal heterogeneity and enclosure temperatures.** The standard deviation of substrate temperatures was compared between enclosure styles and at 0900, 1100, and 1500 using a linear mixed model. At each time point, the substrate temperatures of 12 enclosures of each style were measured. Model predictions were influenced by including enclosure style ($\chi^2_{df=1} = 20.40$, $p < 0.001$), and time of day ($\chi^2_{df=3} = 12.60$, $p = 0.006$), but not their interaction ($\chi^2_{df=3} = 2.79$, $p = 0.42$). Regardless of the time of day, the standard deviation of substrate temperatures was lower in standard enclosures (Untransformed model estimate ± standard error: $\beta_{Standard} = 1.39 \pm 0.07$) compared to naturalistic enclosures ($\beta_{Naturalistic} = 1.86 \pm 0.07$, $p < 0.001$; Fig 8). Furthermore, although the standard deviation of substrate temperatures tended to

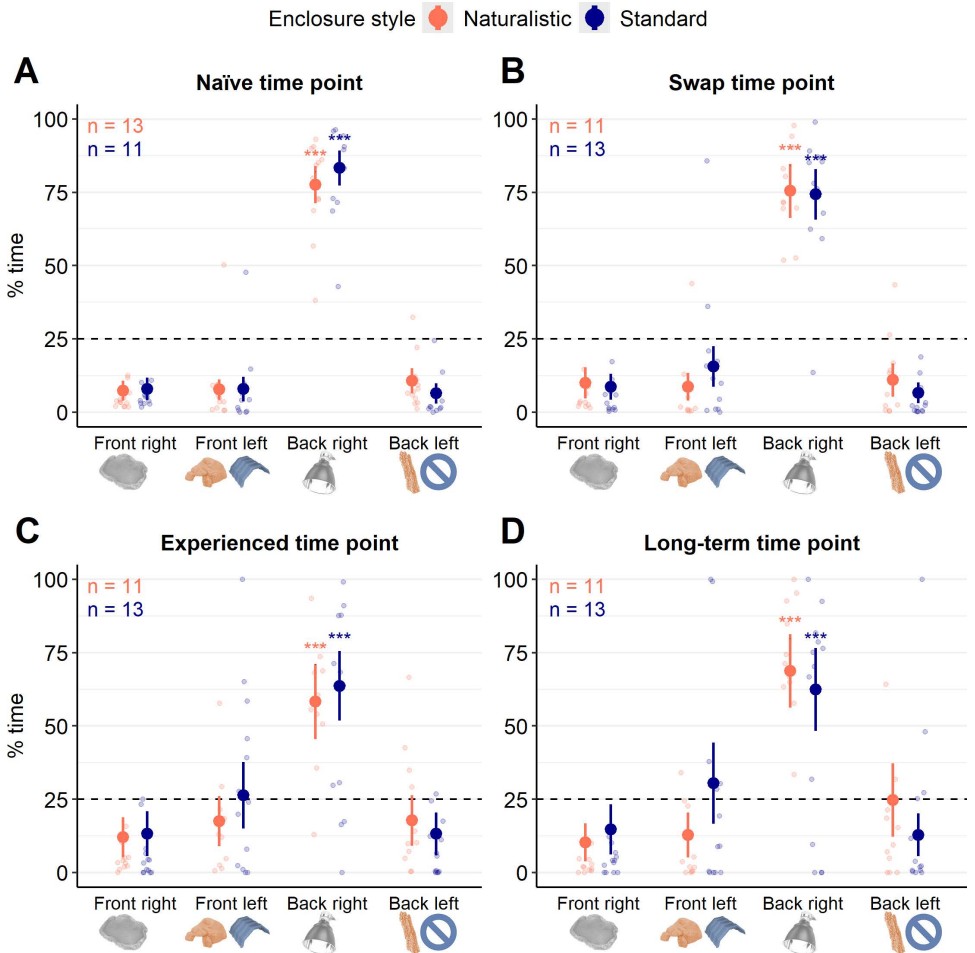

**Fig 7. Enclosure use patterns in Pogona vitticeps housed in naturalistic- or standard-style enclosures.** Percent of a 12-hour day that *Pogona vitticeps* spent in one of four quadrants in either naturalistic (orange) or standard (blue) enclosures at 4 different points in time. Filled-in dots with lines above and below represent transformed model estimates and 97.5% confidence intervals. Lighter-coloured dots represent individual data points. Images below the x-axis highlight the main or only feature in that quadrant and are colour-coded based on enclosure style where relevant (i.e., in the front left quadrant, only naturalistic enclosures had the leftmost/orange hide; only standard enclosures were empty in the back left quadrant). A dotted horizontal line at 25% is included to visualize the amount of time lizards would spend in each quadrant if their behaviour was evenly distributed. Generalized linear mixed models were used to assess time in each quadrant between enclosure styles and included a random intercept for lizard ID. Enclosure style did not influence time in any quadrants at any time points ($p > 0.025$). At each time point, lizards spent the majority of their time in the back right quadrant (colour-coded ***$p < 0.001$).

increase throughout the day, only images taken at 0900 ($\beta_{0900} = 1.49 \pm 0.07$) and 1500 ($\beta_{1500} = 1.72 \pm 0.07$, $p = 0.006$) differed significantly after p-values were adjusted (Fig 8).

At the time the thermal images were taken, basking tiles were removed in 1 standard style and 1 naturalistic style enclosure to reduce wear on the nails of the lizards living in these enclosures. Therefore, the final sample size of this analysis was 22, with 11 enclosures of each style represented at all time points. Tile temperatures were assessed using averages, which were calculated using the average of all visible temperature values recorded on the tile (i.e., within the area of interest), not including the lizard itself. As expected, average tile temperatures fell within acceptable ranges at all times of day and for both enclosure styles; across all time points and regardless of enclosure style, average tile temperatures ranged between 29.78°C and 41.21°C. (Fig 9).

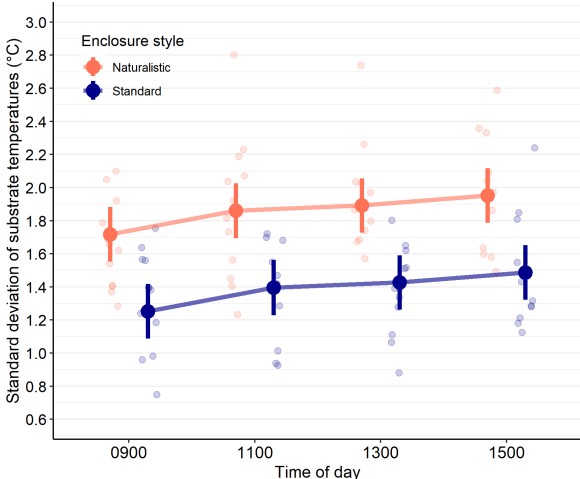

**Fig 8. Naturalistic-style enclosures provide greater variability in substrate temperatures compared to standard-style enclosures.** Standard deviation of substrate temperatures in either naturalistic (orange) or standard (blue) enclosures at 4 different times of day. Filled-in dots with lines above and below represent model estimates and 95% confidence intervals; lighter-coloured dots represent individual data points from enclosures. At each time of day, the substrate from 12 enclosures of each type were measured. Regardless of the time of day, the standard deviation of substrate temperatures was lower in standard enclosures compared to naturalistic enclosures (Linear mixed model, $p < 0.001$). Although the standard deviation of substrate temperatures tended to increase over time, a significant difference was only detected between 0900 and 1500 ($p = 0.006$).

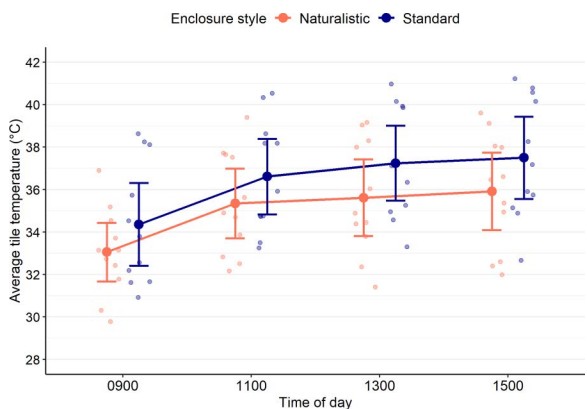

**Fig 9. Average temperature of the tile (basking area) throughout the day in naturalistic- and standard-style enclosures.** Temperatures were measured using thermal imaging and at each time of day, 11 enclosures of each style were measured. Solid dots represent the mean temperature of all enclosures of that style for that time of day. Lines above and below this point represent the 95% confidence interval of this mean. Lighter-coloured dots represent individual within each enclosure.

**Heterophil to lymphocyte (H:L) ratios.** H:L ratios were better explained by a model that included an interaction between enclosure style and sex compared to a model including only additive effects of each factor ($\chi^2_{df=1} = 6.49$, $p = 0.01$). Comparisons within an enclosure style revealed that male lizards in standard enclosures (sample size, median, confidence interval, untransformed model estimates ± standard error: N = 3, 0.10, 0.09–0.13, $\beta_{Male,\ Standard} = -2.25 \pm 0.29$) had lower H:L ratios than female lizards in standard enclosures (N = 9, 0.25, 0.11–0.52, $\beta_{Female,\ Standard} = -1.30 \pm 0.17$; $p = 0.011$). Comparisons within sexes found that female lizards in naturalistic enclosures (N = 4, 0.13, 0.10–0.17, $\beta_{Female,\ Naturalistic} = -2.00 \pm 0.25$) had lower H:L ratios than females in standard enclosures (N = 9, 0.25, 0.11–0.52, $\beta_{Female,\ Standard} = -1.30 \pm 0.17$; $p = 0.032$) (Table 2 and Fig 10).

**Table 2. Values and comparisons of heterophil to lymphocyte (H:L) ratios for *Pogona vitticeps* in naturalistic or standard enclosures.**

| Main effects | | | | | | | | | |
|---|---|---|---|---|---|---|---|---|---|
| Enclosure style | Sex | N | Median | 95% CI | Estimate ± Std. Error | VS. Female | Adj. p-value | VS. Ntrl. | Adj. p-value |
| Naturalistic | Female | 4 | 0.13 | 0.10 - 0.17 | −2.00 ± 0.25 | | | | |
| | Male | 7 | 0.13 | 0.05 - 0.37 | −1.72 ± 0.19 | −0.29 | 0.375 | | |
| Standard | Female | 9 | 0.25 | 0.11 - 0.52 | −1.30 ± 0.17 | | | −0.70 | **0.032** |
| | Male | 3 | 0.10 | 0.09 - 0.13 | −2.25 ± 0.29 | 0.95 | **0.011** | 0.53 | 0.143 |
| Model information | | | | | | | | | |
| | Marginal $R^2$ | | 0.371 | | | | | | |
| | Formula | | HLRatio ~ EnclosureStyle * Sex | | | | | | |
| | Distribution | | Gamma (log link) | | | | | | |

Medians, confidence intervals, model estimates, model standard errors, contrasts between fixed factors, and p-values of associated contrasts are provided. Lizards lived in each enclosure style for at least 600 days (~19 months). Blood cells were identified and counted by an individual blind to the lizard's enclosure style and sex. The glm function with a gamma distribution and log link was used to analyze data, and estimates provided are untransformed. P-values are adjusted according to the Bonferroni method.

Ntrl., Naturalistic enclosure style; HLRatio, Heterophil to lymphocyte ratio; EnclosureStyle, whether the lizard was living in a naturalistic or standard-style enclosure.

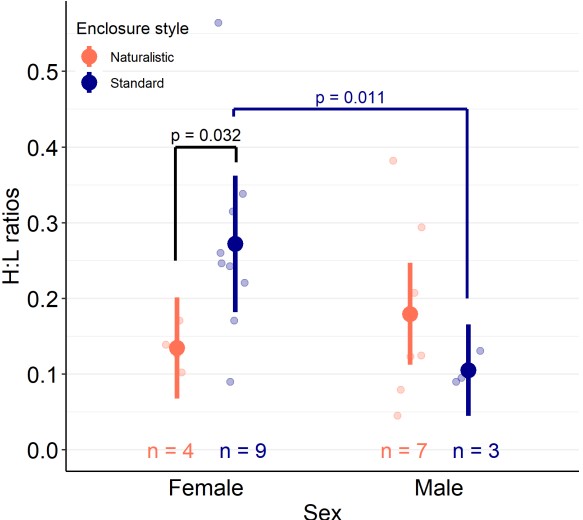

**Fig 10. Heterophil to lymphocyte (H:L) ratios of male or female Pogona vitticeps in naturalistic- or standard-style enclosures.** Filled-in dots with lines above and below represent transformed model estimates and 95% confidence intervals. Lighter-coloured dots represent individual data points. Colour-coded text above the x-axis provides sample sizes for each group. Blood cells were counted by an individual blind to the lizard's sex and enclosure style, and lizards had lived in a particular enclosure style for at least 600 days (~19 months) prior to collection of blood samples. Ratios were analyzed using a generalized linear model with an interaction between enclosure style and sex. Within female lizards, those living in naturalistic enclosures had lower H:L ratios than those in standard enclosures (*p* = 0.032). Within standard enclosures, male lizards had lower H:L ratios than females (*p* = 0.011).

## Discussion

There was no strong evidence that naturalistic enclosures better accommodated a lizard's behaviour or that lizards living in naturalistic enclosures had better welfare. Although naturalistic enclosures had superior thermal heterogeneity, because enclosure style failed to influence enclosure use patterns and only influenced the amount of time spent inactive when enclosure styles were swapped, evidence that naturalistic enclosures could better accommodate a lizard's behaviour is

tenuous. Evidence that naturalistic enclosures improved the lizard's welfare was also equivocal; heterophil to lymphocyte (H:L) ratios were only lower for female lizards living in naturalistic enclosures, and the probability of adopting relaxed postures (inactive and leg stretched, ILS) or performing stress-related behaviour (interacting with barriers, IWB) was similar regardless of enclosure style. Indeed, contrary to the hypotheses and predictions, an influence of sex was detected more often than enclosure style.

Enclosure style only influenced the amount of time spent inactive when lizards were observed within 1 week of their enclosure styles being swapped; at this time point, lizards that moved from naturalistic enclosures into standard enclosures were inactive for a longer amount of time than those moved from standard enclosures into naturalistic ones (Fig 4B). This may indicate that lizards being deprived of naturalistic resources were experiencing stress, providing indirect support for the prediction that naturalistic enclosures could better accommodate the lizards' behaviour. Suddenly losing additional or naturalistic resources has been documented as a potential stressor in other species; fecal glucocorticoids increased after climbing resources were removed from green iguana enclosures [25], and when American mink were moved out of enclosures that facilitated a variety of natural behaviours, the amount of time they spent inactive increased [82]. However, considering that enclosure style failed to influence inactivity at all other time points, this could also be evidence that rearing conditions influence how *P. vitticeps* respond to novelty (half of the lizards lived in standard-style enclosures until enclosure styles were swapped at 1 year of age; see section titled "Enclosure styles" in methods and Table 1). Indeed, it is well-known that enrichment can influence neophobia, anxiety, and cognition in a variety of vertebrates [83–85] including snakes [28,86,87]. For example, as has been seen in rats [84], exposure to enrichment can increase the size certain brain regions [87], reduce habituation time [86], and may improve learning or reduce neophobia [28] in snakes. Regardless, either interpretation must be taken with caution; the size of the observed effect was relatively small, and the relationship between inactivity and stress can be ambiguous [88], especially without concomitant indicators of stress [10,89].

Compared to wild *P. vitticeps*, the amount of inactivity observed was consistently greater in captive lizards. Furthermore, although it was not directly analyzed, there was also a trend of increasing inactivity for all lizards as they matured (Fig 4). This, combined with the equivocal influence of enclosure style on inactivity, fail to support the hypothesis that naturalistic enclosures would better facilitate a lizard's behaviours, but the magnitude of this failing is difficult to discern. Inactivity is a critical component of behavioural thermoregulation for ectotherms, and because thermoregulation is critical for survival, it is likely that inactivity related to thermoregulation is enjoyable [43,90]. Exposure to UV itself could also be pleasurable, as research in endotherms has found [88], and there is evidence that *P. vitticeps* can detect UV [91]. Furthermore, *P. vitticeps* are relatively inactive even in the wild; in some seasons, they move as little as 2 hours per day and may move fewer than 10 minutes per hour [37]. For these reasons, excessive inactivity seems probable for *P. vitticeps* in captivity and therefore may not represent a significant restriction on the lizard's agency. However, even if it is preferred or chosen, excessive inactivity necessarily limits the competence (i.e., the cognitive and behavioural experiences) of a lizard by occupying the majority of their day, which itself can hinder agency [14]. Excessive inactivity can also impair a lizard's health (e.g., obesity, constipation; [6,7,45]). Although it is unclear the degree to which the inactivity observed reflects a restriction in the lizard's agency, these results demonstrate that structural complexity alone was unable to influence the amount of time lizards spent inactive, and concomitantly naturalistic enclosures could not better accommodate the lizard's behaviour.

Regardless of the enclosure style, lizards spent the majority of their day in the back right quadrant of their enclosure throughout the observation period. The main feature of this area was the lamp and tile, which together provided a basking spot, but lizards could also occupy this quadrant by hiding underneath the tile. It is unlikely that the bias for the back right quadrant reflects inadequate basking temperatures; thermal images taken of enclosures demonstrate that basking temperatures always approximated the temperatures at which wild *P. vitticeps* are active and around which captive *P. vitticeps* actively thermoregulate (i.e., between 20°C and 40°C [37,42,48]) (Fig 9). Interestingly, although comparisons between

time points were not directly investigated, the occupation of the front and back left quadrant – in which a naturalistic hide and cork bark were found for naturalistic enclosures, or a paper hide and barren area were found in standard enclosures – seems to increase slightly at later time points (Fig 7C, D). Furthermore, although the difference was not significant, lizards in naturalistic enclosures tended to spend more time in the back left quadrant (which had cork bark) and less time in the front left quadrant (which had the naturalistic hide) compared to standard-housed lizards, who had either a barren area or a paper hide in these quadrants, respectively (Fig 7). This suggests that lizards prefer furnishings that provide an opportunity to climb (i.e., cork bark) over a barren area. Perhaps this is not unexpected as *P. vitticeps* are semi-arboreal and often found on elevated perches in the wild [48,92]. Regardless of these trends, the absence of differences observed in enclosure use patterns between enclosure styles necessitates the conclusion that naturalistic enclosures were at least as effective at accommodating a lizard's behaviour as standard enclosures, failing to support the hypothesis that naturalistic-style enclosures would better facilitate a lizard's agency.

Although the behavioural data failed to support the hypothesis that naturalistic enclosures would better facilitate lizard agency, there was evidence that the substrate of naturalistic enclosures provided better thermal heterogeneity, which may better accommodate behavioural thermoregulation [41]. Because the variation in substrate temperatures was higher in naturalistic enclosures at all times of day (Fig 8), these enclosures provided lizards a greater variety of species-relevant temperatures, which can increase the complexity of the enclosure and may ultimately better facilitate their agency. However, future research must examine if such thermal heterogeneity is actually able to influence body temperatures or other aspects of *P. vitticeps* thermoregulation to support these results.

There was also a general lack of behavioural evidence to support that lizards in naturalistic enclosures experienced better welfare. Lizards in naturalistic enclosures adopted postures likely related to relaxation (inactive and leg stretched, ILS) similarly as often as lizards in standard enclosures. Furthermore, lizards performed behaviours related to escape (interacting with barriers, IWB) for similar amounts of time in both enclosure styles. Only sex influenced the performance of IWB, and this influence was minimal and observed rarely (Fig 6F). Perhaps the lack of influence observed in both behaviours may be partially related to their rarity; ILS and IWB were not observed in all lizards and were typically performed for short periods of time. At later time points, ILS was adopted more often and IWB performed for less time – a trend likely related to the increase in inactivity observed as lizards aged (Fig 4). Although enclosure style failed to influence the performance of ILS, this posture could offer useful insight into lizard welfare for future research, as it is easy to identify and has been described in other species [93]. Furthermore, it is reasonable to assume that ILS is related to relaxation because it is only performed in potentially low-cost situations (i.e., after other bouts of inactivity and ceasing upon being disturbed by external stimuli) and does not appear to confer a thermoregulatory benefit [43]. Identifying behaviours such as ILS that may indicate positive affective states will be vital to better understand reptile welfare [94], but efforts to validate the relationship between ILS and positive affective states is critical to avoid relying on assumptions [10]. In future research, the rarity of ILS and IWB must be considered if they are to be used as potential indicators of affective states; for example, these behaviours may be most effectively measured by observing animals for their entire active period several days in a row to ensure that all instances the behaviour are recorded.

Long-term experience with an enclosure style did influence heterophil to lymphocyte (H:L) ratios, but only for female lizards. Female lizards in naturalistic enclosures had lower H:L ratios than those in standard enclosures, potentially indicating lower chronic stress [44]. In fact, the H:L ratios of standard-housed female lizards was relatively high compared to most other groups (Fig 10). Because blood samples were taken in the spring of 2024, the H:L ratios observed in this group may correlate with the greater amount of IWB typically observed in females during spring [40]. If so, this may be a worthwhile avenue for future research – because all female lizards performed IWB more in spring regardless of their enclosure style [40], could H:L ratios reveal that female lizards in standard enclosures experience greater stress than those in naturalistic-style enclosures, belying the similar amount of IWB they both perform? These findings highlight the importance of considering all potential motivations for a behaviour and, where possible, combining physiological and

behavioural measurements to understand animal welfare. Unfortunately, however, the small sample size of each sex within an enclosure style should promote reluctance when making any definite conclusions about the relationship between naturalistic enclosures and chronic stress for the lizards observed in this study. Furthermore, lizards in standard-style enclosures may have experienced higher stress due to the higher frequency of cage changes they experience, and not due to their enclosure style *per se*, as this has been highlighted as a potential stressor in snakes [95]. However, if this was a stressor for *P. vitticeps*, it is unclear why male lizards would not be influenced, as male *P. vitticeps* are more likely to defend territories in the wild [37], and therefore should be more disturbed by the removal of scent markings.

Overall, we found little evidence that naturalistic enclosures better facilitated a lizard's agency and that lizards in naturalistic enclosures experienced better welfare. All behavioural outcomes (inactivity, enclosure use, ILS, IWB) were similar between enclosure styles and only a moderate influence of sex was detected at certain points in time. However, the substrate of naturalistic enclosures offered better thermal heterogeneity, and H:L ratios were lower in naturalistic-housed compared to standard-housed female lizards. There are a number of potential explanations for the essentially equivocal effect of enclosure design.

Firstly, the absence of significant differences between enclosure styles could be because naturalistic enclosures were not perceived as different from standard enclosures. Essentially, although naturalistic and standard enclosures had different furnishings, these differences were not meaningful to the lizards, causing them to perceive the enclosure styles as similar. To investigate this further, a preference test could be used to examine how lizards perceive the furnishings in each enclosure style [96].

Secondly, the extended time frame over which data were collected may have introduced significant variation in the lizard's behaviour due to season and age, obscuring any potential influence of enclosure style. The extended time frame was originally chosen because it would be most relevant to companion reptiles in captivity and because previous research often examined the influence of enclosure styles after relatively short periods of exposure (e.g., 1–2 weeks to approximately 30 days [22,31,97]). However, as lizards had to be acquired as juveniles to achieve the necessary sample size, data collection began before lizards had reached maturity. In addition, seasonal patterns in behaviour were not anticipated, as the photoperiod and temperatures lizards experienced were held consistent throughout the year. Furthermore, the ability for each enclosure style to accommodate a lizard's behaviour may have changed between seasons and as the lizards matured, as age and season would likely influence the lizard's motivations and needs. Regardless, future research aiming to understand the long-term influence of enclosure style should carefully consider these factors, and data collection should be done several times per individual at each time point to better capture behaviour.

Thirdly, and perhaps most likely, the relatively small size of enclosures could have limited the possibility for naturalistic enclosures to facilitate lizard agency and influence welfare. Research has demonstrated that enclosure size can influence activity levels [23,29,32,98] and enclosure use patterns [34] – although enclosure size is often confounded with additional structural complexity in this research [23,29,34,98]. Enclosure size has been emphasised as one of the most critical aspects of an enclosure because small enclosures are often incapable of offering complexity that is meaningful, useful, or relevant to the animals, regardless of the furnishings that are provided [41]. Indeed, perhaps the occupancy of the quadrant with the cork bark failed to reach significance because perches were too short and thus their perceived value was limited, as wild *Pogona* have been found perching between 20 and 900 cm above the substrate [48,99]. Enclosure sizes would also be perceived as smaller as lizards grew, compounding this issue over time.

Finally, it is also possible that facilitating the agency and assuring the welfare of *P. vitticeps* is relatively straightforward – in other words, the absence of differences is because both enclosure styles were adequate. In the wild, *Pogona* species are relatively inactive and their movement has been observed as direct and purposeful, likely based on the resources available [37,39,99,100], suggesting they rarely roam aimlessly. Therefore, the high amounts of inactivity and skewed enclosure use patterns may be evidence of the preferred state of *P. vitticeps* in captivity; perhaps effective enclosures are simply those that provide a few hiding spots and optimal temperatures. However, the performance of IWB does evince

that captivity imposes at least some restrictions on lizard behaviour, even if these restrictions are not severe, as this behaviour likely indicates that the lizard is motivated to escape the enclosure [40]. Troublingly, reptile owners consider IWB as common or "normal" as non-repetitive activity [101], and our results demonstrate that IWB is frequently observed as the primary form of activity. Accommodating the lizard's agency during their few bouts of activity may therefore improve their welfare, even if lizards prefer to spend most of their day inactive. For example, because *P. vitticeps* are opportunistic predators [102], foraging toys could facilitate agency by simulating the exploitation of found food sources and allowing them to perform more naturalistic eating behaviours (e.g., finding "ant holes" with a large amount of slowly-released insects, foraging balls stuffed with vegetation that must be torn off; [99,100]). Motivations to roam large distances, find mates, or defecate away from home areas could be accommodated even in relatively small enclosures by adding tunnels, satellite enclosures, or treadmills, especially if combined with regularly changing the furnishings of the home cage. Indeed, because the movement of wild *Pogona* species has been observed as rare but purposeful, enclosures that aim to facilitate these occasional motivated bouts of movement may be particularly effective at accommodating behaviour. The trends observed herein suggest that facilitating the agency of *P. vitticeps* may be relatively straightforward, even if greater effort is needed to determine how best to accommodate their sporadic, motivated bouts of activity.

In conclusion, we found that the behaviour of lizards living in enclosures with naturalistic furnishings (i.e., cork bark, loose substrate, extra hiding spaces) was similar to those living in more barren enclosures. Lizards in both enclosure styles spent the majority of their day inactive, did not occupy the areas of the enclosure evenly, adopted postures potentially related to relaxation similarly as often, and performed escape-related behaviours for similar amounts of time. Except for the amount of time spent inactive immediately after their enclosure styles were swapped, the lizard's behaviours were similar even after short (~1 week) and long (140 + days) periods of time in an enclosure style. However, the substrate of naturalistic enclosures offered better thermal heterogeneity, potentially better accommodating their behavioural thermoregulation. Furthermore, after spending almost 2 years in an enclosure style, the H:L ratio was higher for female lizards living in standard enclosures compared to those in naturalistic enclosures, although the sample size for this finding was relatively small. These results demonstrate that naturalistic furnishings alone cannot accommodate a lizard's behaviour, and consequently that the welfare of lizards in both enclosure styles was similar. It is unclear if enclosure design failed to influence the lizards because of how lizards perceived the enclosures, because of the long time frame over which data were collected, because the small size of enclosures rendered both styles inadequate, or because lizards exhibit few behaviours requiring complex enclosures and therefore both styles were adequate.

## Supporting information

**S1 Appendix. Additional details about the camera's blind spot in video recordings.**
(DOCX)

**S2 Appendix. Additional details about analysis of heterophil to lymphocyte ratios.**
(ZIP)

## Acknowledgments

This research would not have been possible without the dedication and hard work of many people. In particular, we would like to thank the diligent efforts of the Animal Care Services staff at Brock University who collected blood samples and effectively safeguarded the health and welfare of the lizards every day throughout the period of data collection. We would also like to thank the many additional personnel who, over the years, assisted with daily care: Jane Oleksiw, Wynne Reichheld, Paige Au, Eleni Tzavelas, Theresa Muraca, and Jackson Rawes. We would also like to thank the Brock Machine Shop, who built the stands used to secure cameras in the enclosures and who built (and maintained) the enclosures themselves. We would also like to thank the lizards themselves, as their behaviour was (most of the time)

enlightening. We acknowledge that data collection took place on the traditional territory of the Haudenosaunee and Anishinaabe peoples and is within the land protected by the Dish with One Spoon Wampum Agreement. The data in this article were gathered as a part of M. D.'s PhD thesis and as a part of N. L. B.'s NSERC Undergraduate Student Research Award.

## Author contributions

**Conceptualization:** Melanie Denommé.

**Data curation:** Melanie Denommé, Natalie L. Bakker, Glenn J. Tattersall.

**Formal analysis:** Melanie Denommé, Natalie L. Bakker, Glenn J. Tattersall.

**Funding acquisition:** Melanie Denommé, Natalie L. Bakker, Glenn J. Tattersall.

**Investigation:** Melanie Denommé, Natalie L. Bakker.

**Methodology:** Melanie Denommé.

**Project administration:** Glenn J. Tattersall.

**Supervision:** Glenn J. Tattersall.

**Validation:** Glenn J. Tattersall.

**Visualization:** Melanie Denommé.

**Writing – original draft:** Melanie Denommé.

**Writing – review & editing:** Melanie Denommé, Glenn J. Tattersall.

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
