## [Decision Letter · Decision Letter 0]

15 Apr 2025

PONE-D-25-16117Influence of enclosure design on the behaviour and welfare of *Pogona vitticeps*PLOS ONE

Dear Dr. Denommé,

Thank you for submitting your manuscript to PLOS ONE. After careful consideration, we feel that it has merit but does not fully meet PLOS ONE’s publication criteria as it currently stands. Therefore, we invite you to submit a revised version of the manuscript that addresses the points raised during the review process.

Thank you for submitting this interesting manuscript; this is potentially a valuable behavioural study with some useful implications for both zoos and private keepers.We received two detailed reviews, both of which indicated that your work had promise. However, both noted that revisions are required, especially in terms of the methods. Please provide more detail pertaining to the enclosure design, as this is important in terms of repeatability. Further detail on experimental procedures is also important here.

We look forward to receiving your revised manuscript.

Kind regards,

James Edward Brereton, MSc

Academic Editor

PLOS ONE

“The data in this article were gathered as a part of M.D.’s PhD thesis. M.D. was supported by a Natural Sciences and Engineering Council (NSERC) Postgraduate Scholarship-Doctoral (PGS D-580167-2023). The data in this article were also gathered as a part of N.L.B.'s NSERC Undergraduate Student Research Award (USRA) (USRA - 592895 - 2024). The research was funded by an NSERC of Canada grant to G.J.T. (RGPIN-2020-05089).”

Reviewers' comments:

Reviewer's Responses to Questions

**Comments to the Author**

1. Is the manuscript technically sound, and do the data support the conclusions?

Reviewer #1: Yes

Reviewer #2: Yes

2. Has the statistical analysis been performed appropriately and rigorously? 

Reviewer #1: Yes

Reviewer #2: Yes

3. Have the authors made all data underlying the findings in their manuscript fully available?

Reviewer #1: Yes

Reviewer #2: Yes

4. Is the manuscript presented in an intelligible fashion and written in standard English?

Reviewer #1: Yes

Reviewer #2: Yes

5. Review Comments to the Author

Reviewer #1: Dear authors, I would like to congratulate you on your work, which is very interesting and relevant. I really enjoyed reading it. However, I have a few suggestions that could make the manuscript even better. In general, the points for improvement are:

1 - There is no information about the biology and behavior of Pogona in the introduction. I suggest inserting a paragraph about this. One idea would be to add the objectives of the study on line 86.

2 - The differences between the two types of enclosure. For me, the so-called natural enclosure didn't differ much from the so-called standard one.

3 - Where was the study that evaluated the inactivity of pogona in the wild carried out? Was it a review? Was it an experiment? Should this study be considered the standard for the species? In the discussion you put some sentences about how pogonas are not very active in the wild. I think this should come up in the discussion as a possible point to take into account when looking at the results.

Please find my comments and suggestions below:

Line 53: (validated, i.e. [10]). I think this needs to be rewritten to make the text clearer. Perhaps, instead of citing the reference, cite some of the validated methods in the sentence.

Lines 86-88: This sentence about gender could be moved to the methodology (suddenly in the data analysis section). But, if you prefer to keep it here in the introduction, my suggestion is that you put why you don't expect a different response from the sex in relation to the type of enclosure.

Lines 98-99: cite references to Pogona's natural behavior.

Lines 105-108: Here I think it depends. For example, if inactivity drops significantly while the diversity of positive behaviors increases? Would this change in the level of inactivity be considered bad? Probably not.

Lines 108-113: Likewise here, it depends. Exacerbated use of an area can be related to what resource that area provides, and that's not necessarily a bad thing. In fact, you found this in your work. The most used area was the lamp area and the structure where the animals could climb. So I think these statements should be more lenient.

Line 118: But wouldn't inactivity be used as a measure of low well-being or of the enclosure's failure to accommodate the lizards' needs? After seeing the results, I'd modify it here and say that you took two measures of inactivity (you explain this later in the paper, because ILS was used in a particular context due to the low number of records).

Line 186: Were the lizards captured and handled for this purpose? The answer is yes, as you'll find out later in the paper. From the results, do you think this might have had an influence on the parameters observed?

Lines 214-221: What behavioral data collection method was used? Focal Instantaneous with registers every 2 seconds?

Line 254: Why was the head used instead of the body? For example, if a pogona had its whole body in quadrant a and its head in quadrant b, was it considered to be in quadrant b?

Fig. 2 caption: Enter the information about the water basin, which was located in two quadrants.

Line 592: change Sample size for sample size

Lines 639-640: Did the rearing conditions vary between individuals? If so, how? If not, explain what you mean here.

Lines 641-642: How enrichment influence these cited characteristics? Explain more and give examples.

Lines 740-745: Most of the parameters analyzed did not differ between the two types of environment, but those that did resulted in better living conditions for the animals in the naturalistic enclosures. Wouldn't this in itself support the idea that naturalistic enclosures would be better, even without changing several parameters?

Lines 747-752: That was my main concern with the study. Was the so-called naturalistic enclosure really naturalistic? I didn't see it that way either. I really enjoyed seeing this paragraph here. My suggestion is that you post ideas on how to better structure an enclosure to the point where it is truly naturalistic. For example, if the animal is semi-arboreal, why aren't there structures like this in the enclosures?

Reviewer #2: This study investigates how enclosure design influences the behavior and welfare of bearded dragons (Pogona vitticeps) in artificial environments. The research compares naturalistic (complex) enclosures with standard (simple) enclosures to determine whether more complex environments improve the welfare of these lizards.

The manuscript is well written, well supported by literature, and presents highly relevant data for the management of Pogona vitticeps. However, one point draws my attention, and I would like more detailed clarification.

- How was the 28-day interval defined for blood collection in establishing the H:L ratio? What reference was used to establish this interval? If the response to environmental changes occurred acutely within a shorter time frame, the authors missed this important moment. Could the authors discuss the sensitivity of the H:L ratio in Pogona vitticeps and the possible implications of an inadequate collection interval, especially for the detection of acute changes?

- Another point that needs adjustment is the definition of a naturalistic enclosure. The modified enclosure cannot be considered naturalistic under any circumstances, as only a few physical structures and a substrate different from the initial phase were introduced.

6. PLOS authors have the option to publish the peer review history of their article (what does this mean? ). If published, this will include your full peer review and any attached files.

**Do you want your identity to be public for this peer review?** For information about this choice, including consent withdrawal, please see our Privacy Policy .

Reviewer #1: No

Reviewer #2: **Yes: ** Cristiane Schilbach Pizzutto

---

## [Author Response · Author response to Decision Letter 0]

1 May 2025

General comments:

Upon review, we realized that for the influence of sex on IWB at the swap point, the direction was incorrectly stated in the results (line 500) and correctly stated in the figure caption. We have now fixed this error.

Also, in this response document, we have occasionally copied text from the original manuscript to effectively highlight any edits we have made. However, as a result, any citations from this copied text are no longer in their original format; i.e., they are provided as names, rather than numbers.

New financial disclosure statement:

The data in this article were gathered as a part of M.D.’s PhD thesis. M.D. was supported by a Natural Sciences and Engineering Council (NSERC) Postgraduate Scholarship-Doctoral (PGS D-580167-2023). The data in this article were also gathered as a part of N.L.B.'s NSERC Undergraduate Student Research Award (USRA) (USRA - 592895 - 2024). The research was funded by an NSERC of Canada grant to G.J.T. (RGPIN-2020-05089). The funders had no role in study design, data collection and analysis, decision to publish, or preparation of the manuscript.

Responses to reviewers:

Reviewer #1: Dear authors, I would like to congratulate you on your work, which is very interesting and relevant. I really enjoyed reading it. However, I have a few suggestions that could make the manuscript even better. In general, the points for improvement are:

1 - There is no information about the biology and behavior of Pogona in the introduction. I suggest inserting a paragraph about this. One idea would be to add the objectives of the study on line 86.

Thank you for this comment; including such detail would better highlight the rationale for several parts of the study. Therefore, we have added a brief mention of wild P. vitticeps behaviour on line 102 (new text underlined below):

Although P. vitticeps are largely inactive in the wild (Bernich et al., 2022; Wild et al., 2022), too little or too much time inactive could indicate that the enclosure is failing to accommodate their needs or behaviours in some way; for example, lizards may bask excessively if the temperatures provided are inadequate or perform repetitive behaviours incessantly in attempts to escape (see Denommé & Tattersall, 2025).

To keep the primary focus of the manuscript on assessing the behaviour and welfare of P. vitticeps, a full paragraph about the natural history of P. vitticeps has been added to the methods, as the first paragraph of the “Animals & husbandry” section.

2 - The differences between the two types of enclosure. For me, the so-called natural enclosure didn't differ much from the so-called standard one.

We chose to label the “naturalistic” enclosure as naturalistic for several reasons. Primarily, “naturalistic” best described the enclosure’s appearance compared to the standard enclosures and also captured the intended outcome of the modifications in the naturalistic enclosure (i.e., to simulate aspects of the natural environment and promote natural behaviours). Although the naturalistic enclosures may also be described as “complex”, this label would have shifted the focus away from the rationale and intended outcome of the furnishings in the naturalistic enclosures. Indeed, because naturalistic enclosures included only 2 additional furnishings (cork bark with paper hide underneath) but modified 2 others (substrate and the naturalistic hide in the front-left quadrant), the term “complex” may have caused readers to overlook these modifications and focus on the number of furnishings.

Furthermore, the contrast between the standard and naturalistic enclosures may effectively replicate the conditions provided to pet reptiles in captivity. Reptiles are often provided, at minimum, a hide and a warm/basking spot (representing the “standard” enclosure), but loose substrate and climbing apparatuses may be less common depending on the species (Cargill et al., 2022; D’Cruze et al., 2020; Howell et al., 2022; Howell & Bennett, 2017). Therefore, our enclosure designs were chosen to have the broadest potential impact, as the pet trade represents a large portion of the reptiles currently kept in captivity (although P. vitticeps are also becoming an increasingly popular model organism for certain research; Fenk et al., 2024). Both enclosure designs were also approved by our local Animal Care Committee and represent aspects that may be feasible in a research environment.

The label “naturalistic” was also chosen as the label “enriched” may have been misleading. In the literature, “enriched” will typically (and should exclusively; Olsson & Dahlborn, 2002) refer to modifications that improve the animal’s welfare in some way, shape, or form (Ratuski & Weary, 2022). As we had no evidence that any of the resources provided to lizards improved the lizard’s well-being beyond the basic requirements, the label “enriched” was avoided. This is also why, throughout the article, we referred to the items in the enclosures as “resources” or “furnishings” rather than “enrichments” (a similar convention has been used in other literature; e.g., Cait et al., 2024).

Therefore, although we agree with the reviewer that the naturalistic enclosures did not closely replicate the lizard’s natural environment, these enclosures can still be viewed as “naturalistic” when compared to the standard enclosures. And, as this manuscript focused only on comparisons between these two enclosure styles, the label “naturalistic” was arguably the most accurate in the context of our experiment. Regardless, to make it clearer that we did not intend for “naturalistic” enclosures to fully replicate the natural environment, we have modified the following sentence in the introduction (new text underlined below):

Because naturalistic enclosures will include additional furnishings and attempt to mimic aspects of the natural environment, we expect these enclosures will better facilitate a lizard’s agency and that the welfare of lizards in these enclosures will be better than those in standard enclosures; we also expect this influence to be evident regardless of the order in which enclosure styles are experienced, and that an enclosure style’s influence will increase the longer it is inhabited.

References cited in response:

Cait, J., Winder, C. B., & Mason, G. J. (2024). How much “enrichment” is enough for laboratory rodents? A systematic review and meta-analysis re-assessing the impact of well-resourced cages on morbidity and mortality. Applied Animal Behaviour Science, 278, 106361. https://doi.org/10.1016/j.applanim.2024.106361

Cargill, B., Benato, L., & Rooney, N. (2022). A survey exploring the impact of housing and husbandry on pet snake welfare. Animal Welfare, 31(2), 193–208. https://doi.org/10.7120/09627286.31.2.004

D’Cruze, N., Paterson, S., Green, J., Megson, D., Warwick, C., Coulthard, E., Norrey, J., Auliya, M., & Carder, G. (2020). Dropping the Ball? The Welfare of Ball Pythons Traded in the EU and North America. Animals, 10(3), 413. https://doi.org/10.3390/ani10030413

Howell, T. J., & Bennett, P. C. (2017). Despite their best efforts, pet lizard owners in Victoria, Australia, are not fully compliant with lizard care guidelines and may not meet all lizard welfare needs. Journal of Veterinary Behavior, 21, 26–37. https://doi.org/10.1016/j.jveb.2017.07.005

Howell, T. J., Warwick, C., & Bennett, P. (2022). Pet management practices of frog and turtle owners in Victoria, Australia. Veterinary Record, 191(12), e2180. https://doi.org/10.1002/vetr.2180

Olsson, I. A. S., & Dahlborn, K. (2002). Improving housing conditions for laboratory mice: A review of “environmental enrichment.” Laboratory Animals, 36(3), 243–270. https://doi.org/10.1258/002367702320162379

Ratuski, A. S., & Weary, D. M. (2022). Environmental Enrichment for Rats and Mice Housed in Laboratories: A Metareview. Animals, 12(4), Article 4. https://doi.org/10.3390/ani12040414

3 - Where was the study that evaluated the inactivity of pogona in the wild carried out? Was it a review? Was it an experiment? Should this study be considered the standard for the species? In the discussion you put some sentences about how pogonas are not very active in the wild. I think this should come up in the discussion as a possible point to take into account when looking at the results.

In this manuscript, we cite several different articles to support the general inactivity of Pogona in the wild: (Bernich et al., 2022; Thompson & Thompson, 2003; Wotherspoon, 2007). Upon review, we realized we failed to cite one important paper, however; (Wild et al., 2022). This paper has now been included wherever we have stated that P. vitticeps are largely inactive in the wild.

This research is all primary and, collectively, examined the behaviour of 230 P. vitticeps, 19 P. minor, and 22 P. barbata in the wild. P. minor and P. barbata were tracked either by using a spool-and-thread technique (Thompson & Thompson, 2003; Wotherspoon, 2007) and P. vitticeps were tracked using radiotelemetry and GPS tracking (Bernich et al., 2022; Wild et al., 2022). Consequently, the method each article used to estimate activity or movement rates differed. As we are unfamiliar with the techniques used to estimate activity levels in the wild, we are unable to comment on what the standard method should be; however, the relatively consistent estimates of inactivity produced by these variable techniques arguably bolsters the reliability of each article’s findings. Furthermore, the relative inactivity of P. vitticeps in the wild was mentioned in our original discussion (see, e.g., lines 779-786, 791-792).

For these reasons, we have not elected to add new details to our manuscript.

References cited in response:

Bernich, A., Maute, K., Contador-Kelsall, I. C., Story, P. G., Hose, G. C., & French, K. (2022). Space use and daily movement patterns in an arid zone agamid lizard. Wildlife Research, 49(6), 557–570. https://doi.org/10.1071/WR20152

Thompson, S. A., & Thompson, G. G. (2003). The western bearded dragon, Pogona minor (Squamata: Agamidae): An early lizard coloniser of rehabilitated areas. Journal of the Royal Society of Western Australia, 86, 1–6.

Wild, K. H., Roe, J. H., Schwanz, L., Georges, A., & Sarre, S. D. (2022). Evolutionary stability inferred for a free ranging lizard with sex‐reversal. Molecular Ecology, 31(8), 2281–2292. https://doi.org/10.1111/mec.16404

Wotherspoon, A. D. (2007). Ecology and management of Eastern Bearded Dragon Pogona barbata. [PhD Thesis]. University of Western Sydney.

Please find my comments and suggestions below:

Line 53: (validated, i.e. [10]). I think this needs to be rewritten to make the text clearer. Perhaps, instead of citing the reference, cite some of the validated methods in the sentence.

This is a great point; it would be judicious to make the process of validating indicators of animal welfare more commonly known. We have added a sentence in these brackets to briefly describe the process from Browning, 2023 (original reference #10).

New text underlined below:

However, little research exists about reptile behaviour compared to other vertebrates (Bonnet et al., 2002; Rosenthal et al., 2017), and there is a considerable lack of validated welfare indicators for reptiles (i.e., validated by repeated testing with multiple other indicators that should all relate to the same state of welfare; (Browning, 2023)), ultimately making it difficult to determine what husbandry practices may be adequate.

Lines 86-88: This sentence about gender could be moved to the methodology (suddenly in the data analysis section). But, if you prefer to keep it here in the introduction, my suggestion is that you put why you don't expect a different response from the sex in relation to the type of enclosure.

Thank you for highlighting this. As suggested, we have modified the mention of sex at this point and in the methods.

We have modified the fourth paragraph of the introduction in the following manner (new text underlined and any text that has been deleted is struck out):

However, because the timing and order by which animals are exposed to enrichment can influence its efficacy (e.g., (Kalliokoski et al., 2012; Nagabaskaran et al., 2022)), we will investigate each hypothesis at multiple points in time, before and after lizards have experienced both naturalistic and non-naturalistic conditions. In addition, because sex can influence the outcomes that will be measured [37,38], it will be included during analysis, but we do not expect that enclosure style will differently influence lizards based on their sex. Because naturalistic enclosures will include additional furnishings and attempt to mimic aspects of the natural environment, we expect these enclosures will better facilitate a lizard’s agency and that the welfare of lizards in these enclosures will be better than those in standard enclosures; we also expect this influence to be evident regardless of the order in which enclosure styles are experienced, and that an enclosure style’s influence will increase the longer it is inhabited. In addition, although sex is known to influence some of the outcomes that will be measured (Bernich et al., 2022; Howard & Jaensch, 2021), enclosure style should not influence lizards differently based on their sex; because we expect that naturalistic enclosures will better facilitate agency, the lizard’s behaviour should be accommodated and their welfare should be improved regardless of the potentially unique motivations of each sex.

We have also modified the “Data analysis” section, 3 paragraphs in to the “General methods for all analyses” section in the following manner (new text underlined):

Because sex could influence the outcomes measured (Bernich et al., 2022; Howard & Jaensch, 2021), it was included during analysis; however, we did not have a priori reasons to expect that sex would influence the potential effects of enclosure style. Likelihood ratio tests were used to identify which factors (e.g., enclosure style, sex, etc.) in what combination (e.g., interaction, additive, separate) most influenced model fits and thereby identify which models would be further investigated. First, likelihood…

Lines 98-99: cite references to Pogona's natural behavior.

Please see our response to the first comment where we have now provided additional information on the biology of Pogona.

Lines 105-108: Here I think it depends. For example, if inactivity drops significantly while the diversity of positive behaviors increases? Would this change in the level of inactivity be considered bad? Probably not.

It is often difficult to relate the amount of time spent inactive to a definitive affective state, whether positive or negative. For this reason, we argued that large deviations from what wild lizards perform could highlight some issue, implying that spending an extreme amount of time inactive is the issue – whether this means that the lizard spends very little or a significant amount of time inactive. However, because it would be erroneous to assume that we know how long P. vitticeps “should” spend inactive each day, we aimed to use the behaviour of wild P. vitticeps to provide context to our lizard’s behaviours; the reasons for this decision are discussed in the manuscript.

The situation the reviewer has described (where inactivity decreases because the animal is performing more beneficial behaviours) does not refute this argument; in this situation, inactivity has strayed away from an extreme to a more moderate level, as the animal is performing a greater variety of behaviours. Furthermore, because lizards are ectothermic and must spend at least some portion of their day inactive to bask, it is reasonable to argue that spending very little time inactive is also problematic – this would mean the animal is not effectively thermoregulating, possibly because they are highly motivated to perform stress-related behaviours.

To make our reasoning clearer, we have added slightly more detail for explanation on lines 105-108 (new text underlined below):

“In either case, large deviations (i.e., extreme values) in the amount of time inactive relative to wild P. vitticeps may reflect that the behaviours that captive lizards can perform are restricted and,

---

## [Decision Letter · Decision Letter 1]

12 May 2025

PONE-D-25-16117R1Influence of enclosure design on the behaviour and welfare of *Pogona vitticeps*PLOS ONE

Dear Dr. Denommé,

Thank you for submitting your manuscript to PLOS ONE. After careful consideration, we feel that it has merit but does not fully meet PLOS ONE’s publication criteria as it currently stands. Therefore, we invite you to submit a revised version of the manuscript that addresses the points raised during the review process.

Thank you so much for completing the reviews. Reviewer 1 has highlighted a few very small final edits.==============================

We look forward to receiving your revised manuscript.

Kind regards,

James Edward Brereton, MSc

Academic Editor

PLOS ONE

Journal Requirements:

Reviewers' comments:

Reviewer's Responses to Questions

**Comments to the Author**

1. If the authors have adequately addressed your comments raised in a previous round of review and you feel that this manuscript is now acceptable for publication, you may indicate that here to bypass the “Comments to the Author” section, enter your conflict of interest statement in the “Confidential to Editor” section, and submit your "Accept" recommendation.

Reviewer #1: All comments have been addressed

Reviewer #2: All comments have been addressed

2. Is the manuscript technically sound, and do the data support the conclusions?

Reviewer #1: Yes

Reviewer #2: Yes

3. Has the statistical analysis been performed appropriately and rigorously? 

Reviewer #1: Yes

Reviewer #2: Yes

4. Have the authors made all data underlying the findings in their manuscript fully available?

Reviewer #1: Yes

Reviewer #2: Yes

5. Is the manuscript presented in an intelligible fashion and written in standard English?

Reviewer #1: Yes

Reviewer #2: Yes

6. Review Comments to the Author

Reviewer #1: Dear authors, thank you very much for resubmitting your work.

Congratulations on the revision of the manuscript and thank you for considering my suggestions. I was very pleased with the responses, I agree with them and I approve the changes made to the body of the paper. I believe that the work is clearer for readers. However, I suggest three more small changes to the manuscript:

1 - In the abstract, on lines 21 and 22, I suggest changing “perfectly replicate native” to “that intend to replicate native”. This is based on your answers to the previous questions. As it stands, it looks like you did this in the study, which was not the case. If you don't think this suggestion is ideal, then I suggest you just rewrite this sentence and the next one so that it doesn't look like you're testing a perfectly naturalistic enclosure.

2 - In line 573, replace & with and, just to keep the text standard.

3 - The explanation of why the term naturalistic was used in the reply letter was excellent. My suggestion is to insert it in the manuscript. I suggest you do this in the Enclosure style methodology section. This explanation really clarifies the researchers' thinking and the justification is quite acceptable.

Well, those are my final suggestions. Once again, I congratulate the authors on their excellent work.

Reviewer #2: (No Response)

7. PLOS authors have the option to publish the peer review history of their article (what does this mean? ). If published, this will include your full peer review and any attached files.

**Do you want your identity to be public for this peer review?** For information about this choice, including consent withdrawal, please see our Privacy Policy .

Reviewer #1: No

Reviewer #2: **Yes: ** Cristiane Schilbach Pizzutto

---

## [Author Response · Author response to Decision Letter 1]

13 May 2025

General comments:

Note that, since the manuscript was submitted originally, we have added some new references to the references section. Specifically, we added:

Reference #39: Wild KH, Roe JH, Schwanz L, Georges A, Sarre SD. Evolutionary stability inferred for a free ranging lizard with sex‐reversal. Mol Ecol 2022;31:2281–92. https://doi.org/10.1111/mec.16404.

Reference #46: Fenk LA, Baier F, Laurent G. The bearded dragon Pogona vitticeps. Nat Methods 2024;21:1964–6. https://doi.org/10.1038/s41592-024-02485-2.

Reference #47: Doneley B. CARING FOR THE BEARDED DRAGON. Proceedings of the North American Veterinary Conference, vol. 20, Orlando, Florida, USA: 2006, p. 1607–11.

Reference #49: Read JL. Subhabitat variability: A key to the high reptile diversity in chenopod shrublands. Aust J Ecol 1995;20:494–501. https://doi.org/10.1111/j.1442-9993.1995.tb00568.x.

Reference #50: Stahl SJ. General Husbandry and Captive Propagation of Bearded Dragons, Pogona vitticeps. Bulletin of the Association of Reptilian and Amphibian Veterinarians 1999;9:12–7. https://doi.org/10.5818/1076-3139.9.4.12.

Reference #51: de Vosjoli P, Sommella TM, Mailloux R, Donoghue S, Klingenberg R, Cole J. The Bearded Dragon Manual: Expert Advice for Keeping and Caring for a Healthy Bearded Dragon. 2nd ed. West Sussex, U.K.: Fox Chapel Publishers International Ltd; 2017.

Reference #53: Olsson IAS, Dahlborn K. Improving housing conditions for laboratory mice: a review of “environmental enrichment.” Lab Anim 2002;36:243–70. https://doi.org/10.1258/002367702320162379.

Reference #54: Mendyk RW, Augustine L. Controlled Deprivation and Enrichment. In: Warwick C, Arena PC, Burghardt GM, editors. Health and Welfare of Captive Reptiles, Cham: Springer International Publishing; 2023. https://doi.org/10.1007/978-3-030-86012-7.

Reference #55: Ratuski AS, Weary DM. Environmental Enrichment for Rats and Mice Housed in Laboratories: A Metareview. Animals 2022;12:414. https://doi.org/10.3390/ani12040414.

Reference #56: Cait J, Winder CB, Mason GJ. How much “enrichment” is enough for laboratory rodents? A systematic review and meta-analysis re-assessing the impact of well-resourced cages on morbidity and mortality. Appl Anim Behav Sci 2024;278:106361. https://doi.org/10.1016/j.applanim.2024.106361.

Reference #57: Williams DL, Jackson R. Availability of Information on Reptile Health and Welfare from Stores Selling Reptiles. Open J Vet Med 2016;06:59–67. https://doi.org/10.4236/ojvm.2016.63007.

Reference #58: Cargill B, Benato L, Rooney N. A survey exploring the impact of housing and husbandry on pet snake welfare. Anim Welf 2022;31:193–208. https://doi.org/10.7120/09627286.31.2.004.

Reference #59: D’Cruze N, Paterson S, Green J, Megson D, Warwick C, Coulthard E, et al. Dropping the Ball? The Welfare of Ball Pythons Traded in the EU and North America. Animals 2020;10:413. https://doi.org/10.3390/ani10030413.

Reference #60: Howell TJ, Bennett PC. Despite their best efforts, pet lizard owners in Victoria, Australia, are not fully compliant with lizard care guidelines and may not meet all lizard welfare needs. J Vet Behav 2017;21:26–37. https://doi.org/10.1016/j.jveb.2017.07.005.

Reference #61: Howell TJ, Warwick C, Bennett P. Pet management practices of frog and turtle owners in Victoria, Australia. Vet Rec 2022;191:e2180. https://doi.org/10.1002/vetr.2180.

Reference #84: Simpson J, Kelly JP. The impact of environmental enrichment in laboratory rats—Behavioural and neurochemical aspects. Behav Brain Res 2011;222:246–64. https://doi.org/10.1016/j.bbr.2011.04.002.

Reference #85: Gatto E, Dadda M, Bruzzone M, Chiarello E, De Russi G, Maschio MD, et al. Environmental enrichment decreases anxiety‐like behavior in zebrafish larvae. Dev Psychobiol 2022;64:e22255. https://doi.org/10.1002/dev.22255.

Reference #87: Nagabaskaran G, Moonilal V, Skinner M, Miller N. Environmental Enrichment Increases Brain Volume in Snakes. J Comp Neurol 2025;533:e70043. https://doi.org/10.1002/cne.70043.

In addition, reference #73 in the original manuscript has been moved to reference #48:

Reference #48: MacMillen RE, Augee ML, Ellis BA. Thermal ecology and diet of some xerophilous lizards from western New South Wales. J Arid Environ 1989;16:193–201. https://doi.org/10.1016/S0140-1963(18)31026-7.

Responses to reviewers:

Reviewer #1: Dear authors, thank you very much for resubmitting your work.

Congratulations on the revision of the manuscript and thank you for considering my suggestions. I was very pleased with the responses, I agree with them and I approve the changes made to the body of the paper. I believe that the work is clearer for readers. However, I suggest three more small changes to the manuscript:

1 - In the abstract, on lines 21 and 22, I suggest changing “perfectly replicate native” to “that intend to replicate native”. This is based on your answers to the previous questions. As it stands, it looks like you did this in the study, which was not the case. If you don't think this suggestion is ideal, then I suggest you just rewrite this sentence and the next one so that it doesn't look like you're testing a perfectly naturalistic enclosure.

Thank you for highlighting how this phrasing could be misleading. We have made the requested edit, although we chose slightly different wording (new text underlined below):

However, designing such enclosures can be difficult if little is known about the animal in the wild, and enclosures that aim to replicate natural habitats must still be assessed to ensure their assumed benefits are realized.

2 - In line 573, replace & with and, just to keep the text standard.

We have made the requested edit.

3 - The explanation of why the term naturalistic was used in the reply letter was excellent. My suggestion is to insert it in the manuscript. I suggest you do this in the Enclosure style methodology section. This explanation really clarifies the researchers' thinking and the justification is quite acceptable.

We appreciate the comment. As suggested, we have added 2 paragraphs within the methods justifying our enclosure designs and terminology. These paragraphs appear immediately after the figure 1 caption. These paragraphs heavily resemble our original response, but we have edited them to improve the flow of ideas and remove improper phrasing (i.e., references to “the readers”).

Well, those are my final suggestions. Once again, I congratulate the authors on their excellent work.

Thank you for your revisions as well! We have appreciated your professional tone and insightful comments.

---

## [Editor Report · Decision Letter 2]

15 May 2025

Influence of enclosure design on the behaviour and welfare of *Pogona vitticeps*

PONE-D-25-16117R2

Dear Dr. Denommé,

We’re pleased to inform you that your manuscript has been judged scientifically suitable for publication and will be formally accepted for publication once it meets all outstanding technical requirements.

Kind regards,

James Edward Brereton, MSc

Academic Editor

PLOS ONE
---

## [Editor Report · Acceptance letter]

PONE-D-25-16117R2

PLOS ONE

Dear Dr. Denommé,

I'm pleased to inform you that your manuscript has been deemed suitable for publication in PLOS ONE. Congratulations! Your manuscript is now being handed over to our production team.

Kind regards,

on behalf of

Mr. James Edward Brereton

Academic Editor

PLOS ONE